# Domain Randomization via Entropy Maximization

**Gabriele Tiboni**[1,2,*], **Pascal Klink**[2], **Jan Peters**[2,4,5,6], **Tatiana Tommasi**[1],
**Carlo D'Eramo**[2,3,4], **Georgia Chalvatzaki**[2,4,7]

[1]Department of Control and Computer Engineering, Politecnico di Torino, Italy
[2]Department of Computer Science, Technische Universität Darmstadt, Germany
[3]Center for Artificial Intelligence and Data Science, University of Würzburg, Germany
[4]Hessian Center for Artificial Intelligence (Hessian.AI), Darmstadt, Germany
[5]Centre for Cognitive Science, TU Darmstadt, Germany.
[6]Systems AI for Robot Learning, German Research Center for AI (DFKI)
[7]Center for Mind, Brain and Behavior, Uni. Marburg and JLU Giessen, Germany
[*]Corresponding author: `gabriele.tiboni@polito.it`

## Abstract

Varying dynamics parameters in simulation is a popular Domain Randomization (DR) approach for overcoming the reality gap in Reinforcement Learning (RL). Nevertheless, DR heavily hinges on the choice of the sampling distribution of the dynamics parameters, since high variability is crucial to regularize the agent's behavior but notoriously leads to overly conservative policies when randomizing excessively. In this paper, we propose a novel approach to address sim-to-real transfer, which automatically shapes dynamics distributions during training in simulation without requiring real-world data. We introduce DOmain RAndomization via Entropy MaximizatiON (DORAEMON), a constrained optimization problem that directly maximizes the entropy of the training distribution while retaining generalization capabilities. In achieving this, DORAEMON gradually increases the diversity of sampled dynamics parameters as long as the probability of success of the current policy is sufficiently high. We empirically validate the consistent benefits of DORAEMON in obtaining highly adaptive and generalizable policies, *i.e.* solving the task at hand across the widest range of dynamics parameters, as opposed to representative baselines from the DR literature. Notably, we also demonstrate the Sim2Real applicability of DORAEMON through its successful zero-shot transfer in a robotic manipulation setup under unknown real-world parameters.

## 1 Introduction

The low sample efficiency of Reinforcement Learning (RL) algorithms and the expensive data collection routine on real hardware have limited the development of fully autonomous robots for the real world. In addition, trial and error approaches such as RL require random exploration to find optimal policies, which raises crucial safety concerns for applications in robotics (Kober et al., 2013). For these reasons, training in simulation is a promising alternative direction for data-driven robot learning, providing a safe way to collect data with minimal costs (Zhao et al., 2020). However, the discrepancy between the simulated environment and the real setup—commonly referred to as the *reality gap*—ultimately hinders policy transfer (Valassakis et al., 2020). Sim-to-real methods attempt to close the reality gap in various ways, *e.g.*, inferring simulator parameters from real data (Zhu et al., 2018), randomizing parameters to encourage policy robustness (Peng et al., 2018; Antonova et al., 2017), or combining both approaches (Tiboni et al., 2023; Chebotar et al., 2019).

Domain Randomization (DR) is the sim-to-real approach that varies simulator parameters according to a given distribution at training time (Muratore et al., 2022b), effectively inducing an additional source of stochasticity over the environment dynamics. Intuitively, DR trades optimality for robustness (Josifovski et al., 2022): excessively high randomization leads to over-regularized policies

that are no longer able to reach optimal performance; on the other hand, unnecessarily narrow distributions may result in policies that fail to generalize. Notably, to address the artificially induced non-stationarity by DR, we can cast the problem into a partially-observable one, and employ non-Markovian (*e.g.*, history-dependent) policies to learn adaptive behaviors over the varied dynamics parameters (Chen et al., 2022). However, DR still requires tedious manual tuning to get candidate training distributions which would generalize best to the real world (Vuong et al., 2019b).

In response to these issues, various methods in the literature propose to automatize DR for achieving better zero-shot transfer, in the absence of real-world data. Yu et al. (2018) work around the partially-observable setting by training dynamics-conditioned policies, but then rely on a test-time parameters search to find an optimal policy. More recent works suggest gradually guiding the training distribution over time, to maximize the average performance over a fixed reference range of dynamics (Mozian et al., 2020), or over the boundary of the training uniform distribution (Akkaya et al., 2019). While promising, these works require iteratively testing the policy on out-of-distribution dynamics or biasing sampled training data towards the boundary of the current distribution. As a result, considerably more environment interactions than a traditional DR pipeline are needed.

In this paper, we propose a novel method to automatically guide the training dynamics distribution, without requiring real-world data. We follow the intuition that better generalization can be achieved with increased diversity of sampled dynamics parameters. Therefore, we propose our method for DOmain RAndomization via Entropy MaximizatiON (DORAEMON), which directly maximizes the entropy of the distribution at training time, *i.e.*, gradually solving the task over more diverse environment dynamics. Crucially, we constrain the growth of entropy by the probability of success of the current policy, to ensure fair convergence along the process and avoid a performance collapse from excessive randomization. DORAEMON only requires a notion of task success to be provided given the task at hand, such as a task-solved return threshold or a success indicator function based on a custom metric—*e.g.*, distance from a goal location. This way, the DR sampling distribution can be automatically widened to reach the maximum entropy value such that policy success still occurs with arbitrarily high probability. As a result, the complexity of the problem shifts from tuning a number of dynamics parameter distributions, to simply defining a binary rule that determines whether trajectories are considered a success or not. In particular, we found our approach to be consistently more sample efficient than Mozian et al. (2020) and Akkaya et al. (2019), as no additional environment interactions are needed for updating the distribution besides episodes naturally experienced at training time. Our thorough experimental evaluation in simulation demonstrates the capabilities of DORAEMON to solve common benchmark tasks in a much wider range of dynamics parameters, hence achieving better generalization than DR baselines. Finally, we show the remarkable Sim2Real transferability of policies trained through DORAEMON, enabling a 7-DoF robotic arm to push a box under unknown center of mass, weight, and contact dynamics.

## 2 RELATED WORK

Domain Randomization (DR) has been widely investigated in recent years as a method to overcome the reality gap (Muratore et al., 2022b; Zhao et al., 2020). Although initially studied in the context of randomization for visual properties of simulators (Tobin et al., 2017), our work deals with the non-stationarity of dynamics distributions—*e.g.* varying friction coefficients, masses. This formulation—a.k.a. *Dynamics Randomization*—generally aims at learning policies that are robust to changes in dynamics, and demonstrated a number of success stories of zero-shot sim-to-real transfer for deep DRL (Antonova et al., 2017; Tan et al., 2018; Ding et al., 2021). Interestingly, more recent works employ history-dependent policies to allow for implicit system identification and adaptive behavior, which demonstrated superior performance (Peng et al., 2018; Akkaya et al., 2019). This is permitted by the natural implementation of DR where new dynamics parameters are sampled at the start of each training episode, effectively inducing latent MDPs (Chen et al., 2022).

Despite better training convergence through adaptivity, training policies with DR still requires considerable manual tuning (Vuong et al., 2019a). For this reason, finding automatic ways to obtain suitable training DR distributions would be a promising solution. Simulator-based inference methods allow the collection of real-world data for estimating posterior distributions over dynamics parameters, which are ultimately used for DR (Tiboni et al., 2023; Chebotar et al., 2019; Tsai et al., 2021; Muratore et al., 2022a). These methods are often referred to as Adaptive Domain Randomization,

and generally depart from the inference objective used and/or assumptions over the data collection routine (Tiboni et al., 2022), *i.e.* whether only fixed off-policy trajectories are given vs. iteratively allowing policy evaluation in the real world.

A complementary research direction investigates the sim-to-real transfer challenge in the absence of real data, as in the scope of this work. Muratore et al. (2019) proposed a novel stopping criterion for training RL policies in simulation by assessing generalizability within the DR distribution with additional policy evaluations. Similarly, Mozian et al. (2020) suggest directly optimizing for average performance over a reference dynamics range but instead propose to train on a moving distribution which is guided accordingly. Notably, these methods require frequently evaluating policies through Monte-Carlo rollouts, resulting in poor data efficiency. Alternatively, Mehta et al. (2020) attempt to deviate from the usual DR formulation and design a learned sampling strategy guided by policy search itself: a sampler policy is rewarded for choosing dynamics parameters where the task policy behaves noticeably more differently than normal. Note how the aforementioned approaches still crucially rely on a reference DR distribution to be provided at training time, as it is necessary for assessing generalizability. In contrast, Akkaya et al. (2019) promises a fully automated algorithm for DR, by leveraging the performance of the policy at the boundaries of a uniform distribution as a proxy for assessing generalizability in each dynamics dimension separately. As a result, however, this approach is by definition confined to uniform distributions and significantly biases policy training on dynamics parameters collected at the boundaries (*e.g.* half the training time in the original implementation). In comparison, our work relaxes both these assumptions while addressing the same general setting—no reference DR distribution must be defined. Interestingly, Akkaya et al. (2019) also demonstrate that the gradual growth of the sampling distribution yields superior overall performance w.r.t. a fixed target, likely due to an effect of curriculum learning (Bengio et al., 2009).

In the context of RL, curriculum learning has shown promising results by adjusting the task difficulty over time (Baranes & Oudeyer, 2010; Fournier et al., 2018; Cho et al., 2023)–*i.e.* the curriculum. This line of work resembles our setting in that the training environment is non-stationary and guided to maximally benefit the agent. In particular, *self-paced learning* approaches recently addressed the problem of automatically generating a curriculum towards a final target task by ensuring a sufficiently high degree of performance along the process (Klink et al., 2020b;a; 2022; Koprulu & Topcu, 2023). Although related, our work deviates from self-paced curriculum learning methods by maximizing the entropy of a distribution over hidden dynamics parameters to facilitate the generalization of learned policies, instead of making an agent proficient on a specified distribution of a target task—See Appendix B for a thorough comparison of the two problem settings.

## 3 BACKGROUND

Consider the discrete-time dynamical system described by a Markov Decision Process (MDP) $\mathcal{M}$, with state space $\mathcal{S}$, action space $\mathcal{A}$, initial state distribution $\mu \colon \mathcal{S} \to \mathbb{R}^+$, transition dynamics distribution $\mathcal{P} \colon \mathcal{S} \times \mathcal{A} \times \mathcal{S} \to \mathbb{R}^+$ and reward function $\mathcal{R} \colon \mathcal{S} \times \mathcal{A} \times \mathcal{S} \to \mathbb{R}$. At each time $t$, the *environment* $\mathcal{M}$ evolves according to the current state $s_t \in \mathcal{S}$ and action $a_t \in \mathcal{A}$ taken by an *agent*, *i.e.* the decision maker, with initial state $s_0 \sim \mu(\cdot)$. As a result of the state transition, a scalar reward signal $r_t = \mathcal{R}(s_t, a_t, s_{t+1})$ is returned. We then formulate the DR problem by modeling the simulator as a set $\mathcal{U}$ of MDPs with tunable latent dynamics parameters $\xi \in \Xi \subseteq \mathbb{R}^{n_\xi}$. Each MDP $\mathcal{M}_\xi \in \mathcal{U}$ shares the same state space, action space, and reward function, but differs by its associated transition dynamics $\mathcal{P}_\xi(s_{t+1}|s_t, a_t)$. Dynamics parameters $\xi$ are generally assumed to be random variables distributed according to a parametric distribution $\nu_\phi \colon \Xi \to \mathbb{R}^+$, parametrized by $\phi \in \Phi \subseteq \mathbb{R}^{n_\phi}$. Such distribution defines the sampling probability of environments $\mathcal{M}_\xi$ at training time. In other words, the agent is iteratively presented with a random sample $\mathcal{M}_\xi$ and can learn from experience by collecting *trajectories* $\tau = \{(s_t, a_t, r_t, s_{t+1})\}_{t=0}^{T-1} \in \mathcal{T}$, encoding the resulting state-action-reward tuples visited. Under this formulation, DR for RL addresses the problem of finding an optimal stochastic policy $\pi_\theta^*(a_t|s_t)$ such that the expected (discounted) cumulative reward is maximized, while acting over an *unknown* distribution of MDPs $\mathcal{M}_\xi$, induced by $\xi \sim \nu_\phi(\cdot)$:

$$J(\theta, \phi) = \mathbb{E}_{\xi \sim \nu_\phi(\xi)} \left[ \mathbb{E}_{\tau \sim p_\theta(\tau|\xi)} \left[ \sum_{t=0}^{T-1} \gamma^t \mathcal{R}(s_t, a_t, s_{t+1}) \right] \right],$$

$$p_\theta(\tau|\xi) = \mu(s_0) \prod_{t=0}^{T-1} \mathcal{P}_\xi(s_{t+1}|s_t, a_t) \pi_\theta(a_t|s_t)$$

(1)

with discount factor $\gamma \in [0, 1)$. In particular, notice how $\pi_\theta$ is not conditioned on the latent dynamics parameters $\xi$. In turn, this formulation limits the capabilities of the policy to adapt its behavior according to different environment dynamics. As in partial observable environments, we therefore adopt the standard approach of considering a history of transitions $\psi \in \Psi$, where $\Psi$ is the set of all possible histories of transitions of the form $\psi_t = \{s_0, a_0, s_1, a_1, \ldots, a_{t-1}, s_t | s_t \in \mathcal{S}, a_t \in \mathcal{A}\}$, where $s_0 \sim \mu(\cdot)$ and $a_t \sim \pi_\theta(\cdot | \psi_t)$.

## 4 METHOD

Our method learns a policy $\pi_\theta$ proficient on a real-world MDP $\mathcal{M}^*$ by training it purely in simulation on a set of MDPs $\mathcal{M}_\xi \in \mathcal{U}$. Given that we do not know which parameters $\xi^*$ correspond to the real-world MDP $\mathcal{M}^*$, we maximize the chance of good performance in $\mathcal{M}^*$ by training a policy $\pi_\theta$ that generalizes well to the maximum range of environments in $\mathcal{U}$. To effectively measure the degree of generalizability across $\mathcal{U}$, we leverage a notion of success for the underlying trajectories $\tau \in \mathcal{T}$, as it gives a better definition of "acceptable" behaviors across tasks. For example, defining success through task-specific knowledge is often more natural for real-world tasks in robotics, as done in (Florensa et al., 2018; 2017). Thus, without loss of generality, we introduce a simple success indicator function $\sigma : \mathcal{T} \rightarrow \{0, 1\}$, which may be defined through, *e.g.*, distance thresholds from a target goal location, task-specific tolerance to errors, or a lower bound on the expected return. Based on the success indicator $\sigma(\tau)$, the probability of success

$$\mathcal{G}(\theta, \phi) = \mathbb{P}[\sigma(\tau) = 1 | \theta, \phi] \tag{2}$$

assesses the capabilities of the policy $\pi_\theta$ to solve the task in simulation over multiple dynamics $\xi \sim \nu_\phi(\cdot)$. The trajectories $\tau$ are now effectively distributed according to the marginal distribution $p_{\theta,\phi}(\tau) = \int \nu_\phi(\xi) p_\theta(\tau | \xi) d\xi$. Wider dynamics distributions $\nu_\phi(\xi)$ will generally make it harder for the policy to keep a high success rate $\mathcal{G}(\theta, \phi)$. On the other hand, high variability of environments $\mathcal{M}_\xi$ sampled at training time will likely lead to better generalization to the real environment dynamics. Following this intuition, we introduce **Do**main **Ra**ndomization via **E**ntropy **M**aximizati**on** (**DORAEMON**), a constrained optimization problem formulated as

$$\max_{\theta \in \Theta, \phi \in \Phi} \mathcal{H}(\nu_\phi) \quad \text{s.t.} \quad \mathcal{G}(\theta, \phi) \geq \alpha, \tag{3}$$

with $\mathcal{H}(\nu_\phi) = -\mathbb{E}_{\xi \sim \nu_\phi(\xi)}[\log \nu_\phi(\xi)]$ being the (differential) entropy of $\nu_\phi$, and desired in-distribution success rate $\alpha \in [0, 1] \subset \mathbb{R}$.

DORAEMON aims at automatically finding a policy that generalizes well to the widest range of dynamics parameters $\xi$. To this end, we propose maximizing the entropy of the training distribution $\nu_\phi(\xi)$ to gradually increase the diversity of sampled environments, towards assigning equal (uniform) probability density to each $\mathcal{M}_\xi \in \mathcal{U}$. While doing so, we ensure that the policy is still able to retain a desired probability of success $\alpha$, which prevents policy performance from collapsing due to excessive randomization. In general, $\nu_\phi$ might *not* converge to the maximum entropy uniform distribution—referred to as $\nu_{\max}$, assuming bounded support—but, instead, will stop when the probability of success $\alpha$ may no longer be maintained. It is worth noting that $\alpha$ does not directly determine the probability of success $\mathcal{G}(\theta, \nu_{\max})$ of the policy across $\nu_{\max}$. Notably, policies with lower in-distribution success rate $\alpha$ may still generalize better thanks to a higher-entropy training distribution (see success rate vs. entropy trade-off in Sec. 5).

### 4.1 ALGORITHMIC IMPLEMENTATION

The objective in (3) requires the joint optimization of the training distribution $\nu_\phi$ and the policy $\pi_\theta$. In practice, we consider a decoupled optimization of the policy with any RL subroutine of choice, which interleaves dynamics distribution updates. Therefore, we conveniently rewrite (3) as

$$\max_{\phi_{i+1} \in \Phi} \mathcal{H}(\nu_{\phi_{i+1}}) \quad \text{s.t.} \quad \mathcal{G}(\theta_i, \phi_{i+1}) \geq \alpha \quad D_{KL}(\nu_{\phi_{i+1}} \| \nu_{\phi_i}) \leq \epsilon, \tag{4}$$

where $\pi_{\theta_i}$ has been trained on dynamics parameters drawn from $\xi \sim \nu_{\phi_i}(\cdot)$. Importantly, $\pi_{\theta_i}$ is conditioned on a fixed-length history over previous state-action pairs to account for the non-observability of environment parameters $\xi$. In this work, we train policies with Soft Actor-Critic (SAC) (Haarnoja et al., 2018), and additionally condition the critic network with the true dynamics

parameters, as in Peng et al. (2018); Akkaya et al. (2019), to further mitigate the non-stationarity induced by sampling dynamics from $\nu_\phi$. We then constrain the Kullback-Leibler (KL) divergence between subsequent dynamics distribution updates. The resulting trust region around $\nu_{\phi_i}$ prevents abrupt changes and controls the growth of the training distribution during the optimization process. Note that the probability of success $\mathcal{G}(\theta, \phi)$ is equal to $\mathbb{E}_\tau[\sigma(\tau)]$, as $\sigma(\tau)$ is effectively distributed as a Bernoulli random variable. Thus, we can approximate the success rate $\mathcal{G}(\theta_i, \phi_{i+1})$ in (4) through the importance sampling (IS) estimator

$$\hat{\mathcal{G}}(\theta_i, \phi_i, \phi_{i+1}) = \frac{1}{K} \sum_{k=1}^K \frac{\nu_{\phi_{i+1}}(\xi_k)}{\nu_{\phi_i}(\xi_k)} \cdot \mathbb{1}_{\{\tau \in \mathcal{T}: \sigma(\tau)=1\}}(\tau_k). \tag{5}$$

In turn, this allows the optimization problem in (4) to be solved using only the set of data $\{(\xi_k, \tau_k)\}_{k=1}^K$ naturally collected while training the policy $\pi_{\theta_i}$ for K episodes. In principle, one could collect additional Monte-Carlo evaluations of the policy to approximate $\mathcal{G}(\theta_i, \phi_{i+1})$ with the most recent parameters $\theta_{i+1}$. However, recycling training data works sufficiently well throughout our experiments and consequently leads to a more efficient pipeline.

While convenient, approximating the probability of success $\mathcal{G}(\theta_i, \phi_{i+1})$ through IS may lead to over-estimation of the real success rate under the new distribution $\phi_{i+1}$, resulting in a violated constraint when trying to solve (4) at the next iteration. This occurrence may be particularly troublesome for DR settings, as there are no guarantees that a policy may solve the task over the entire set of MDPs $\mathcal{U}$. To account for potential constraint violations, we introduce a back-up optimization problem which attempts to find a new feasible starting solution $\phi_i^B$ within the current trust region, formulated as

$$\phi_i^B = \arg\max_{\phi_i' \in \Phi} \hat{\mathcal{G}}(\theta_i, \phi_i, \phi_i') \text{ s.t. } D_{KL}(\nu_{\phi_i'} \| v_{\phi_i}) \le \epsilon. \tag{6}$$

In simple words, we find a sufficiently close distribution that has maximum (approximated) in-distribution success rate. We observed this addition to be crucial for recovering policy performance by backtracking the distribution entropy, which would otherwise be prevented. The overall practical implementation of DORAEMON is summarized in Algorithm 1.

In this work, we parameterize $\nu_\phi(\xi)$ as uncorrelated univariate Beta distributions, allowing for simpler comparison with methods that require bounded support. In addition, dynamics parameters are often inherently bounded due to physical constraints–such as positive values for masses and friction coefficients–making Beta distributions a reasonable choice. However, the formulation is not restricted to a particular family of parametric distributions or even continuous random variables (See Sec. A.4)–*e.g.*, discrete distributions over object shapes could be used.

## 4.2 TOY PROBLEM

We report a practical example of DORAEMON in a toy problem to convey a clear intuition of our method. Let us consider a frictionless inclined plane, with inclination angle $\omega$ (Fig. 1a). An actuated cart (*i.e.* the agent) is placed on the plane surface and is subject to gravity force $F_g$. The cart's goal is to apply a counter force $a_t \in [-a_{\max}, a_{\max}]$ to fight gravity and balance itself around the center of the plane. In this setting, $\xi := \omega \in [-\frac{\pi}{2}, \frac{\pi}{2}] \subset \mathbb{R}$ is the underlying dynamics parameter for our problem. Therefore, the objective is to learn a dynamics-agnostic policy $\pi_\theta$ which can balance the cart across the widest range of inclinations $\omega$. First, note how the policy benefits from keeping a history of past actions and observations for solving the task (see Sec. 3), as $\omega$ can be implicitly inferred from previous state-transitions. The agent can only successfully solve the task for instances of $\omega$ where gravity has a sufficiently low impact. This represents the general problem of learning under potentially infeasible dynamics in DR settings as the distribution gets wider and wider. Therefore, we use DORAEMON to automatically find the maximum entropy distribution where the task may still be solved with probability $\alpha$ without requiring additional domain knowledge. Notably, the simple task at hand allows us to analytically compute the feasibility boundaries of $\omega$, namely $|\omega| \le \omega_c = \arcsin \frac{a_{\max}}{F_g}$ assuming $0 \le a_{\max} \le F_g$. This makes it easy to assess the capabilities of DORAEMON to solve the task over all parameters across the true, feasible range.

We report the results of policies trained with our method in Fig. 1, together with an illustration of the underlying task. We highlight the effect of the hyperparameter $\alpha$ in shaping the final distribution (colored in dark blue): policies with lower $\alpha$ values encounter infeasible tasks more often while

---

**Algorithm 1:** Domain Randomization via Entropy Maximization (DORAEMON)

---

**input** : Initialize dynamics distribution $\phi_1$, policy parameters $\theta_1$, trust region size $\epsilon$, trajectories per distribution update $K$, number of iterations $M$, success indicator function $\sigma(\tau)$.

**output:** Generalizable policy $\pi_{\theta_M}$

1 **for** $i = 1, \ldots, M$ **do**
2      Sample $K$ dynamics parameters $\{\xi_k\}_{k=1}^{K}$ , $\xi_k \sim \nu_{\phi_i}(\xi)$
3      Collect $K$ associated trajectories $\{\tau_k\}_{k=1}^{K}$ , $\tau_k \sim p_{\theta_i}(\tau | \xi_k)$
4      **Policy update:**
5      Obtain $\theta_{i+1}$ through any RL algorithm on collected data $\{\tau_k\}_{k=1}^{K}$
6      **Dynamics distribution update:**
7      **if** $\hat{\mathcal{G}}(\theta_i, \phi_i, \phi_i) < \alpha$ **then**
8          Obtain $\phi_i^B$ by optimizing (6)
9          **if** $\hat{\mathcal{G}}(\theta_i, \phi_i, \phi_i^B) < \alpha$ **then** **jump to next iteration** with $\phi_{i+1} \leftarrow \phi_i^B$ ;
10          $\phi_i^{\text{start}} \leftarrow \phi_i^B$
11      **else**
12          $\phi_i^{\text{start}} \leftarrow \phi_i$
13      **end**
14      Obtain $\nu_{\phi_{i+1}}$ by optimizing (4) with $\mathcal{G} \approx \hat{\mathcal{G}}(\theta_i, \phi_i^{\text{start}}, \phi_{i+1})$ (Eq. 5)
15 **end**

---

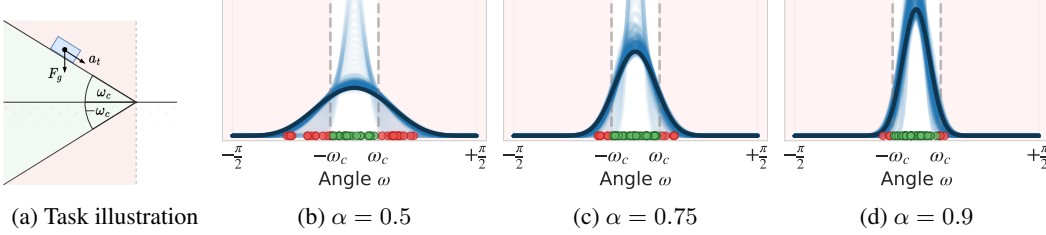

(a) Task illustration      (b) $\alpha = 0.5$      (c) $\alpha = 0.75$      (d) $\alpha = 0.9$

Figure 1: DORAEMON's moving Beta distributions over the plane inclination angle $\omega$, for different values of in-distribution success rate $\alpha$. The converged "max-entropy" distribution is such that the policy can solve the task with probability $\alpha$ (green for success, red for failure). The physically infeasible dynamics region is highlighted with a red background.

training, whereas higher $\alpha$ lead to a more conservative final entropy. For each policy, we finally sample 50 dynamics parameters from the converged final distribution and display them as green dots to indicate success and red for failure. Here, we define a trajectory to be successful if the agent balances the cart around the center of the plane for at least 25 timesteps. Overall, regardless of the in-distribution success rate, all policies learn to solve the task across the widest range of feasible parameters $\omega \in [-\omega_c, \omega_c]$, which is our ultimate goal.

## 5 EXPERIMENTS

### 5.1 BASELINE METHODS

**No-DR.** We refer to policies trained on a single simulator instance with fixed dynamics parameters as No-Domain Randomization (No-DR). This naive approach reflects the baseline capabilities to generalize to the range of environments in $\mathcal{U}$ when no randomization is applied.

**Fixed-DR.** As our method guides the sampling DR distribution over time, we compare it with the simple popular approach of keeping a fixed distribution at training time. We consider a set of parametric MDPs $\mathcal{U}$ with bounded support and train Fixed-DR policies with a uniform distribution that covers the whole range $\mathcal{U}$, *i.e.* the maximum entropy distribution $\nu_{\text{max}}$.

**LSDR.** Mozian et al. (2020) introduce a novel method to guide DR distributions. In contrast to DORAEMON, LSDR requires a reference DR distribution to be given. Then, a (multivariate Gaussian) training distribution $\nu_\phi$ is found such that maximum generalization to the reference distribution is achieved. In our experiments, we set the entropy of the initial distribution to be the same as DORAEMON's, and we set $\nu_{\text{max}}$ as the reference distribution.

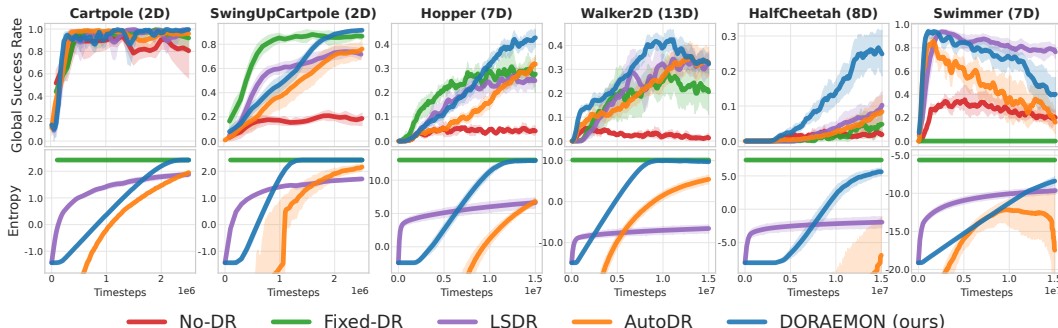

Figure 2: Sim-to-Sim results: global success rate computed on the maximum-entropy uniform distribution (top) and entropy of the current training DR distribution (bottom). The number of randomized parameter dimensions is reported in parenthesis (see Table 2 for details).

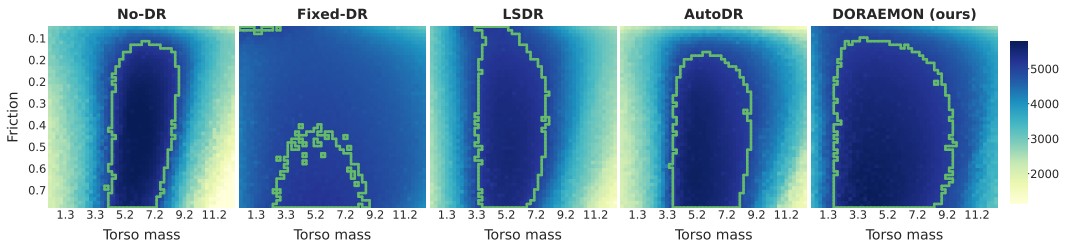

Figure 3: Average test return across 2 dynamics parameters, in the HalfCheetah environment (10 seeds). The green outline highlights the region of cells where a success occurs ($\sigma(\tau) = 1$).

**AutoDR.** Automatic Domain Randomization (AutoDR) by Akkaya et al. (2019) gradually increases the entropy of a uniform DR distribution according to the performance of the policy measured at the boundaries. Analogously to our work, a notion of success is defined through an average return threshold, which determines whether the uniform bounds may be widened. As in the original paper, the sampling distribution $\nu_\phi$ is initialized to approach zero variance and located at the center of $\mathcal{U}$.

## 5.2 SIM-TO-SIM TRANSFER

We conduct a thorough experimental evaluation of DORAEMON on six benchmark tasks in simulation, from the OpenAI Gym (Brockman et al., 2016) MuJoCo environments. We define a specific set of randomized dynamics parameters $\Xi$ tailored to each environment–*e.g.* link masses and sizes, surface friction coefficients–together with an associated success indicator function $\sigma(\tau)$. We define success through a lower bound on the trajectory return, to allow a fair comparison with AutoDR and LSDR which have not been tested before on performance metrics beyond the reward function (see Appendix A for the full details). We then conveniently use $\sigma(\tau)$ to measure the generalization of each method across all MDPs $\in \mathcal{U}$—namely $\mathcal{G}(\cdot, \nu_{max})$—referred to as the *Global Success Rate*.

**Effect of entropy maximization on policy generalization.** The core results of our evaluation are illustrated in Fig 2. During training, we progressively report the entropy of the (moving) distributions and the global success rate of the policies on the maximum entropy $\nu_{max}$ distribution (10 seeds per method). In general, we observe that policies trained with DORAEMON demonstrate a consistent trend of better and/or faster convergence across all tasks. We found LSDR to steadily converge to intermediate entropy values as a consequence of their opt. problem formulated as an M-projection (see Appendix B for details). In contrast, AutoDR was able to regularly increase the training distribution entropy, albeit with considerably higher variance across different seeds. Yet, DORAEMON outperforms AutoDR in all environments. We suspect AutoDR shortcomings are due to inefficient use of training data: collected returns can only be used to update one dimension of the uniform distribution (at most), or even discarded if the performance threshold is not met. Conversely, DORAEMON makes use of all sampled dynamics parameters to update $\nu_{\phi_i}$, acting on all parameter dimensions in $\Phi$ at the same time. Interestingly, the degradation in performance over time in the Walker2D and Swimmer tasks is likely due to the agent's exposure to harder/infeasible parameters, which destabilize training. To mitigate this effect, we track the best-performing policy during

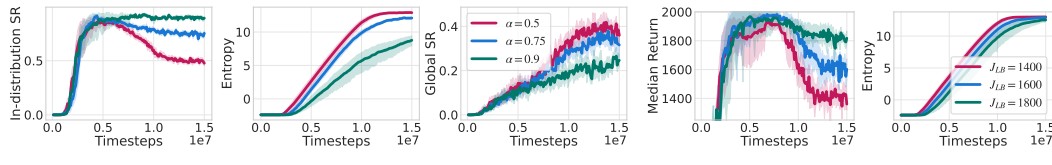

(a) Success Rate (SR) analysis        (b) Performance vs Entropy analysis

Figure 4: Analysis on the impact of the hyperparameter $\alpha$ (a), and of the provided lower bound return threshold $J_{\text{LB}}$ for defining success (b) in the Hopper environment.

training in terms of global success rate. Finally, we report the performance of the best-performing policies on the HalfCheetah environment in Fig. 3, by testing on a discretized 2D-slice of the dynamics space—the remaining parameters are kept fixed at the nominal values. The figure attests the ability of DORAEMON to solve the task over the widest range of dynamics parameters.

**Effect of curriculum on policy training.** Comparing the performances of DORAEMON with Fixed-DR policies in Fig. 2 sheds light on the importance of gradually widening the training distribution, rather than sampling parameters from the maximum entropy one directly. DORAEMON always outperforms Fixed-DR policies—even when it eventually converges to $\nu_{\max}$—likely due to an induced curriculum over dynamics parameters, in line with findings in Akkaya et al. (2019).

**Success rate vs. Entropy trade-off.** The in-distribution success rate $\alpha$ regulates the trade-off between increasing distribution entropy and exposing the agent to dynamics that may not be solved yet. Limiting the distribution entropy to encompass only successful tasks may stabilize training but, on the other hand, prevent generalization to out-of-distribution dynamics that have never been seen. We study this trade-off and report the results for the Hopper environment in Fig. 4a. We find that the value of $\alpha = 0.5$ generalizes sufficiently well—likely due to training on higher entropy values—and selected this value for all our experiments. This way, we assure that a desirably high *median* return is maintained. Notably, the median is not affected by the value of catastrophic returns collected on infeasible dynamics parameters, as a plain average would. This further motivates the use of the success rate as a metric to guide our entropy maximization objective, rather than the average return.

**Performance vs. Entropy trade-off.** While $\alpha$ is a hyperparameter of our method, the success indicator function $\sigma(\tau)$ should be defined through domain knowledge. The algorithm should behave agnostically w.r.t $\sigma(\tau)$, hence tracking the desired success rate $\alpha$ equally well, regardless of the success threshold defined. Fig. 4b illustrates this behavior for the Hopper environment, by setting $\sigma(\tau) = \mathbb{1}_{\{\tau \in \mathcal{T}: J(\tau) \geq J_{\text{LB}}\}}(\tau_k)$, for three performance lower bounds $J_{\text{LB}} = \{1400, 1600, 1800\}$, with fixed $\alpha = 0.5$. We report the *median* performance and study the ability of DORAEMON to collect returns $J(\tau) \geq J_{\text{LB}}$ at least half the time along the process. We observe that the algorithm successfully exploits the success indicator to drop below optimal performance as much as the respective thresholds allow, trading performance for robustness to wider dynamics (*i.e.*, higher entropy).

## 5.3 SIM-TO-REAL TRANSFER

We finally assess the proposed method in obtaining policies that can transfer well to the real world. To this end, we design a novel dynamics-sensitive task tailored to the study of learning generalizable policies across unobservable dynamics. We introduce the *PandaPush* environment, a 7DoF robotic manipulation task with the goal of pushing a square box with unknown center-of-mass towards a target location[1]. We reproduce the real setup in simulation with MuJoCo's physics engine (Todorov et al., 2012) and use DORAEMON to train a single policy that can be successfully deployed over different center-of-mass configurations (see Appendix C for details).

We consider a total number of 17 dynamics parameters to mitigate the reality gap, including box mass (1), surface friction coefficient (1), joint damping and friction (14), and center of mass (1) along the perpendicular axis to the pushing direction. We remark that adding the randomization of more parameters besides the center of mass is crucial to get a smooth deployment of policies in the real world. We report the results of our method compared to the baselines in Tab. 1. Notably, we observe impressive behavior both in simulation and on the real setup when deploying policies trained with DORAEMON (cf. Fig 6).

---

[1]Videos available on the project website https://gabrieletiboni.github.io/doraemon/.

|  |  | No-DR | Fixed-DR | LSDR | AutoDR | DORAEMON (ours) |
|---|---|---|---|---|---|---|
| Sim2Sim | success rate | 15.06% | 0.02% | 37.77% | 30.45% | **66.57%** |
|  | distance to target (cm) | $11.66 \pm 6.55$ | $21.04 \pm 2.70$ | $8.30 \pm 7.38$ | $8.27 \pm 6.39$ | $\mathbf{3.17 \pm 3.04}$ |
| Sim2Real | success rate | 13.33% | 0% | 46.67% | 26.67% | **60%** |
|  | distance to target (cm) | $8.08 \pm 6.28$ | $23.26 \pm 2.25$ | $7.38 \pm 8.69$ | $4.17 \pm 2.07$ | $\mathbf{2.68 \pm 1.01}$ |

Table 1: PandaPush task: success rate and final distance of box w.r.t. goal (cm) tested for the maximum entropy distribution averaged over 10000 rollouts for Sim2Sim, and 30 rollouts for Sim2Real. The task is successfully solved if the agent pushes the box within a 3cm radius of the goal.

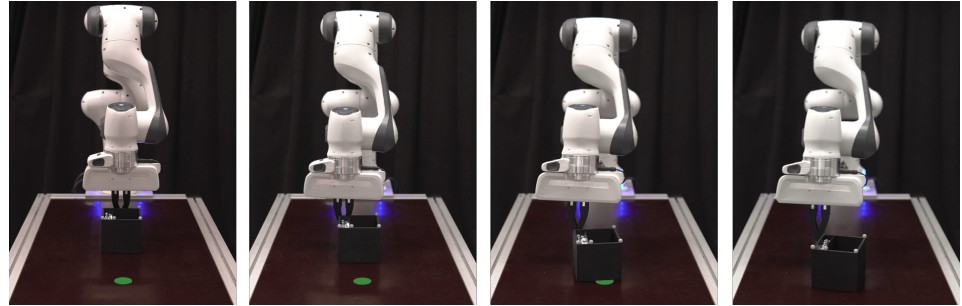

Figure 6: Illustration of a real-world rollout when deploying a policy trained with DORAEMON, at four snapshots during the 6-second trajectory. The history-based policy shows impressive behavior right from the start, as slightly touching the box reveals the shifted center of mass, hence the gripper is promptly moved accordingly (second snapshot figure). The green dot highlights the goal location.

Importantly, we find LSDR to be poorly scalable to our 17-dimensional dynamics space—the computational complexity for approximating $J(\theta, \nu_{\max})$ through Monte-Carlo policy evaluations grows exponentially with the number of randomized dynamics—leading to higher variance across multiple seeds. Analogously, AutoDR shows limited performances, which we attribute to the data-inefficient growth of the uniform distribution, *i.e.* slower pace than DORAEMON. In turn, this prevented AutoDR policies from encompassing enough variability in the dynamic space to generalize well to the real world. Finally, all 10 policies trained on a fixed wide DR distribution (Fixed-DR) are unable to learn any meaningful behavior, highlighting the problem of learning on excessively diverse dynamics.

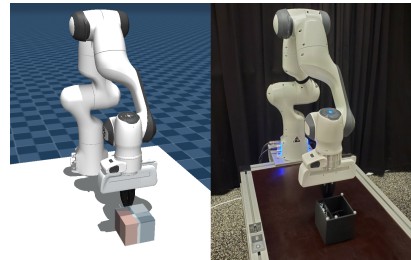

Figure 5: PandaPush setup: the 7DoF robot arm needs to push a box of varying center-of-mass to a desired location.

## 6 CONCLUSION AND DISCUSSION

We presented a novel method for Domain Randomization via Entropy Maximization (DORAEMON), which constitutes a significant step in addressing the sim-to-real transfer gap in Reinforcement Learning (RL). DORAEMON automates the selection of simulator dynamics parameters at training time by progressively widening the sampling distribution, alleviating the need for costly real-world data and manual tuning. Importantly, we constrain the entropy maximization such that sufficiently high policy performance is maintained along the process. In turn, DORAEMON proficiently balances the trade-off between policy convergence and generalization, *i.e.* a common challenge in domain randomization. Our empirical results, both in Sim2Sim control tasks and a versatile Sim2Real manipulation task with a 7DoF robotic arm, showcased the superior performance of DORAEMON w.r.t. representative baselines, underscoring its potential in narrowing the reality gap.

**Limitations.** We suspect DORAEMON performance might suffer from collapsing to some "easy" region of the optimization landscape when backtracking from the current distribution—see Appendix B for details. Adding a KL constraint between the current policy and the best-performing policy found during training could perhaps prevent catastrophic forgetting in these cases. Moreover, if prior knowledge of the dynamics is given, it may be beneficial to bias the growth of the distribution to stay around it, despite being a harder optimization problem to solve.

## ACKNOWLEDGMENTS

This work was funded by the German Federal Ministry of Education and Research (BMBF) (Project: 01IS22078). This work was also funded by Hessian.ai through the project 'The Third Wave of Artificial Intelligence – 3AI' by the Ministry for Science and Arts of the state of Hessen. The authors gratefully acknowledge the scientific support and HPC resources provided by the Erlangen National High Performance Computing Center (NHR@FAU) of the Friedrich-Alexander-Universität Erlangen-Nürnberg (FAU) under the NHR project b187cb. NHR funding is provided by federal and Bavarian state authorities. NHR@FAU hardware is partially funded by the German Research Foundation (DFG) – 440719683. We further acknowledge the support of the European H2020 Elise project (www.elise-ai.eu), for the availability of HPC resources and support. This study was carried out within the FAIR - Future Artificial Intelligence Research and received funding from the European Union Next-GenerationEU (PIANO NAZIONALE DI RIPRESA E RESILIENZA (PNRR) – MISSIONE 4 COMPONENTE 2, INVESTIMENTO 1.3 – D.D. 1555 11/10/2022, PE00000013). This manuscript reflects only the authors' views and opinions, neither the European Union nor the European Commission can be considered responsible for them.

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

|  | Parameters | Boundaries | $J_{\text{LB}}$ threshold |
|---|---|---|---|
| CartPole, SwingUpCartpole (2D) | Gravity (1D) | $[2.39, 17.21]$ | 400, 4200 |
|  | Pole length (1D) | $[0.12, 0.88]$ |  |
| Hopper (7D) | Link masses (4D) | $[0.35, 9.75]$ | 1600 |
|  | Joint damping (3D) | $[0.17, 2.93]$ |  |
|  | Surface Friction (1D) | $[0.17, 2.93]$ |  |
| Walker2D (13D) | Torso mass (1D) | $[0.88, 6.19]$ | 1600 |
|  | Thigh masses (2D) | $[0.98, 6.87]$ |  |
|  | Leg masses (2D) | $[0.68, 4.75]$ |  |
|  | Foot masses (2D) | $[0.74, 5.15]$ |  |
|  | Torso size (1D) | $[0.10, 0.70]$ |  |
|  | Thigh sizes (1D) | $[0.11, 0.79]$ |  |
|  | Leg sizes (1D) | $[0.15, 1.05]$ |  |
|  | Foot sizes (1D) | $[0.05, 0.35]$ |  |
|  | Friction Left Foot (1D) | $[0.48, 3.32]$ |  |
|  | Friction Right Foot (1D) | $[0.22, 1.58]$ |  |
| HalfCheetah (8D) | Torso mass (1D) | $[0.32, 12.4]$ | 5000 |
|  | Back thigh mass (1D) | $[0.08, 2.99]$ |  |
|  | Back shin mass (1D) | $[0.08, 3.08]$ |  |
|  | Back foot mass (1D) | $[0.05, 2.08]$ |  |
|  | Front thigh mass (1D) | $[0.07, 2.78]$ |  |
|  | Front shin mass (1D) | $[0.06, 2.30]$ |  |
|  | Front foot mass (1D) | $[0.04, 1.66]$ |  |
|  | Surface friction (1D) | $[0.02, 0.78]$ |  |
| Swimmer (7D) | Link masses (3D) | $[28.42, 40.70]$ | 112 |
|  | Link sizes (3D) | $[0.08, 0.12]$ |  |
|  | Viscosity coefficient (1D) | $[0.08, 0.12]$ |  |

Table 2: Search space boundaries for all the randomized parameters of the considered simulation tasks, together with the lower bound return threshold used for defining success.

## A   SIM-TO-SIM EXPERIMENTS

Here, we describe the details of the experimental setting carried in simulation, on 6 tasks from the OpenAI gym suite of environments.

### A.1   SIMULATION ENVIRONMENTS SPECIFICATIONS

For each test environment, we select a number of dynamics parameters to formulate the Domain Randomization problem tailored to the properties of the task. The number of randomized parameters ultimately define the optimization space for LSDR, AutoDR and DORAEMON. Moreover, we define finite boundaries for each environment search space, such that a well-defined maximum entropy distribution exists for benchmark purposes. This allows the comparison of our method with Fixed-DR and LSDR—which require a reference target distribution to be defined—and allows dealing with inherently physically bounded parameters (*e.g.* positive values for friction coefficients).

We report the list of randomized properties for each task, together with the search space boundaries considered, in Tab. 2.

In general, we designed the boundaries to be as wide as possible, while being centered in the nominal parameter values and avoiding unstable values—*e.g.* masses or link sizes that are too close to zero. We therefore optimize policies with DORAEMON using Beta distributions $\mathcal{B}e(a, b)$ defined in such ranges, with initial shape parameters $\mathcal{B}e(a = 100, b = 100)$. However, in principle, any distribution may be used. For LSDR, we stick to the original implementation setting which uses Multivariate Gaussian distributions, and set an initial diagonal covariance matrix such that the entropy is equal to the starting distribution entropy of DORAEMON. Note that LSDR required an entropy computation that follows from truncated Gaussian probability density function, as we resample parameters that lie outside of the search space. For optimization stability and consistency across different parameter scales, each dimension is rescaled to the interval $[0, 1]$ at optimization time for all methods. All dis-

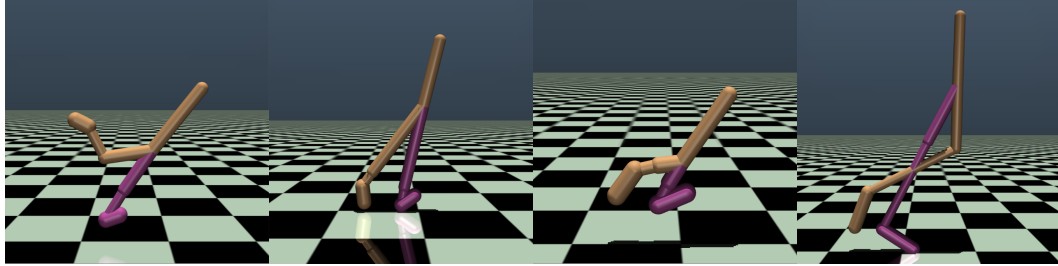

Figure 7: Different random configurations of the Walker2D environment when sampling dynamics parameters from the maximum entropy distribution.

tributions are initialized to the center of the corresponding search space. We illustrate an example of randomized configurations for the Walker2D environment in Fig. 7. Finally, an environment-specific success notion has been defined in order to run AutoDR and DORAEMON, and consequently used to measure the success rate of all methods. As stated in the main manuscript, the success notion can be freely defined given domain knowledge and tolerance to errors. Throughout our sim-to-sim experiments, we define success by means of a simple lower bound on the trajectory return, which we report in Tab. 2. We design threshold values $J_{LB}$ corresponding to acceptably good qualitative results—*e.g.* Cartpole being balanced, Hopper jumping forward)—and such that we roughly stay around $80\%$ of the performance of a policy trained without DR on the nominal parameters.

## A.2    HYPERPARAMETERS CHOICE

We benchmark all methods by training on the same RL subroutine, *i.e.* with SAC, with identical hyperparameters. In particular, policies are given the additional information on the 5 most recent state-action pairs, whereas critic networks are conditioned with the privileged information of the true dynamics parameters, which get rescaled before being fed to the network. We then individually tune the hyperparameters of the considered baselines for a fair comparison: for each method, we perform a grid-search over its hyperparameters to obtain optimal average performance across all tasks; then, we separately tune a single selected hyperparameter per method on each environment individually. In particular, we choose to separately tune $\alpha$ for LSDR, $\Delta$ for AutoDR, and $\epsilon$ for DORAEMON—the notation here is referring to the respective papers notation. Note that these parameters generally regulate the pace of the growing distribution. Fig. 8 demonstrates an illustrative example of the tuning process of $\epsilon$ for DORAEMON. We observe that higher values allow the distribution to grow faster, but may lead to excessive update steps which hinder policy training and generalization. Interestingly, regardless of the choice of the trust region size $\epsilon$, the backup optimization problem in (6) allows the entropy to be adjusted—e.g. slightly reduced—to maintain the desired in-distribution success rate $\alpha = 0.5$. As a side note, this effect is however harder to notice for distributions that are close to the max-entropy uniform state, as even considerable changes to the distribution parameters only marginally affect the entropy in this region—i.e. the derivative of the entropy w.r.t. the Beta parameters $\mathcal{B}e(a, b)$ approaches zero as $(a, b)\rightarrow(1, 1)$. With this in mind, in Fig. 9 we observe an example of noticeably different Beta distributions, despite *seemingly* having the same entropy according to Fig. 8. Exceptionally for LSDR, we also vary the number of Monte Carlo (MC) evaluations in the reference distribution, as it's crucially affected by the dimensionality of the dynamics space. We then set LSDR to collect $20 \cdot n_\xi$ episode returns—where $n_\xi$ is the number of randomized parameters—which is in line with the original paper experimental setting. Importantly, notice that while a higher number of evaluations generally leads to a better approximation, this translates to a much higher number of environment interactions w.r.t. the baselines. In the case of AutoDR, we also have the flexibility to choose a "low" return threshold value $t_L$, used to backtrack the uniform distribution when average boundary performance falls below $t_L$—the analogous mechanism the we induce with our backup optimization problem in (6). In contrast to our implementation, however, this required an additional parameter to be selected for AutoDR: we then set $t_L$ to be half of the return threshold $J_{LB}$ throughout our experiments, as done in the original work (Akkaya et al., 2019).

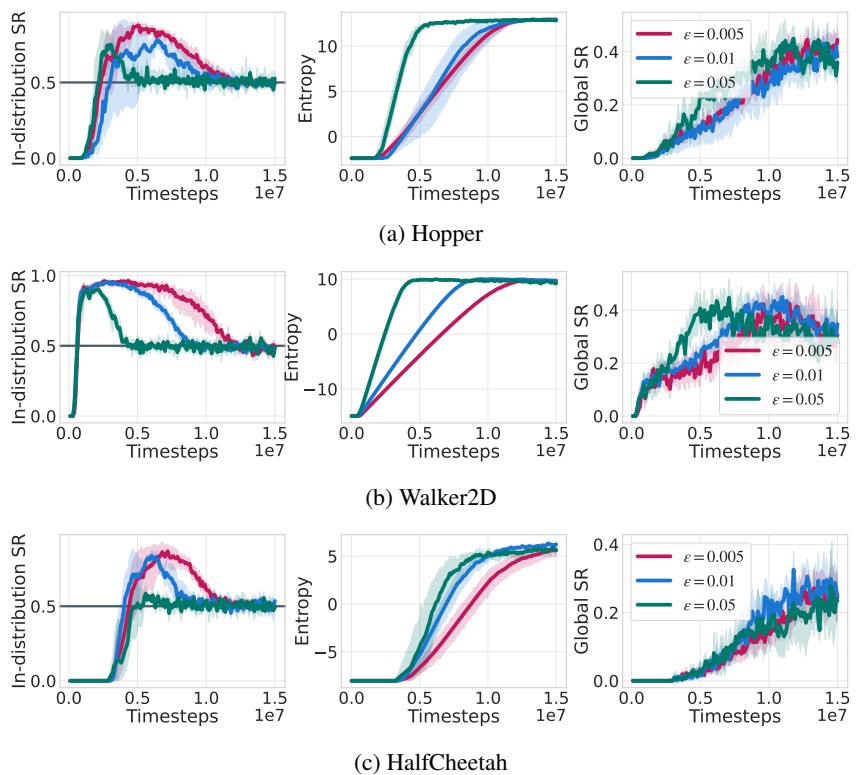

(a) Hopper

(b) Walker2D

(c) HalfCheetah

Figure 8: Sensitivity analysis of the trust region size $\epsilon$ on Hopper (top), Walker2D (middle), and HalfCheetah (bottom) environments, with $\alpha{=}0.5$. DORAEMON is able to consistently maintain the in-distribution success rate even for larger trust region sizes. SR stands for *success rate*.

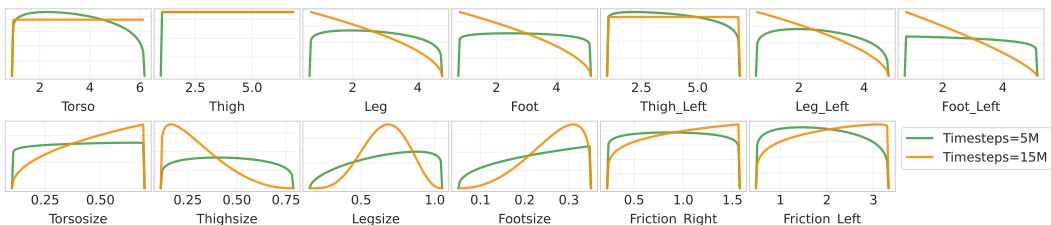

Figure 9: Comparison of Beta distributions at different times during training with DORAEMON on the Walker2D environment with $\epsilon = 0.05$ (see Fig. 8b for comparison). The green Beta distribution obtained after 5M timesteps (entropy=9.93) is adjusted along the process to maintain the desired in-distribution success rate, and eventually converges to the orange distribution after 15M training timesteps (entropy=8.46).

Overall, we select the best configuration found in terms of global success rate. Refer to our public code implementation at https://gabrieletiboni.github.io/doraemon/ for the full reproducibility of our experimental evaluation.

## A.3 SUCCESS RATE VS. ENTROPY TRADE-OFF

This section augments and summarizes the homonymous analysis in the main experimental section (Sec. 5.2) with further experiments and considerations on the trade-off between the hyperparameter $\alpha$ and its effect on policy training and generalization. We illustrate the overall results of this analysis in Fig. 10. To interpret the figures effectively, we shall look at the evolution of in-distribution success rate w.r.t. the growth of entropy, and ultimately how these policies generalize by means of global success rate. Intuitively, a higher value of $\alpha{=}0.9$ constrains DORAEMON to be conservative with

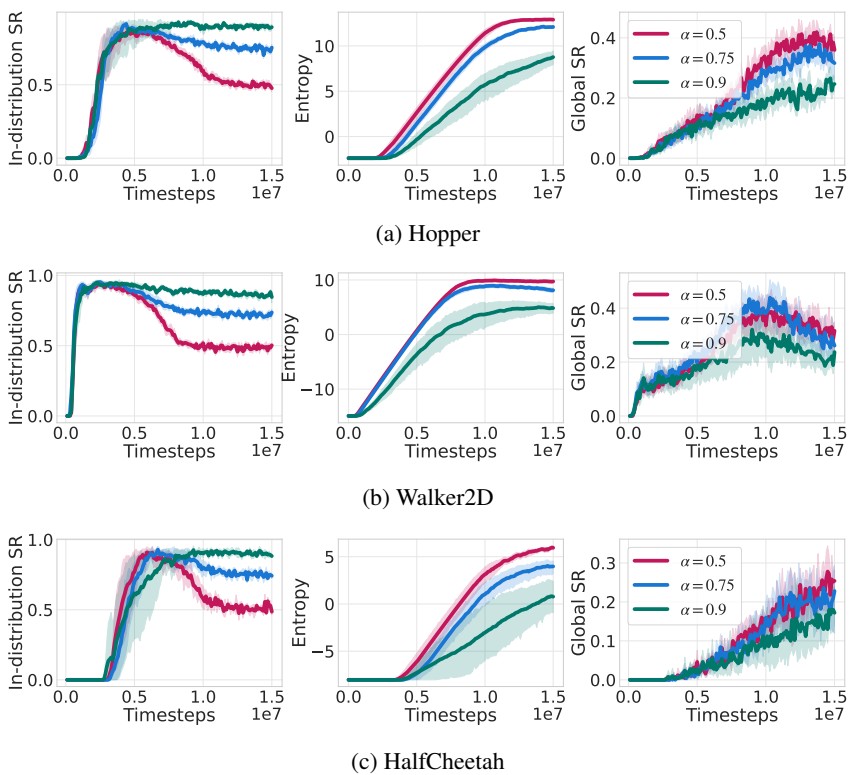

Figure 10: Analysis on the impact of the hyperparameter $\alpha$ on Hopper (top), Walker2D (middle), and HalfCheetah (bottom) environments. The desired in-distribution success rate $\alpha=\{0.5, 0.75, 0.9\}$ is tracked effectively by DORAEMON.

the increase of entropy, leading to policies that likely overfit on in-distribution dynamics and poorly generalize. On the contrary, a less restrictive value $\alpha=0.5$—i.e. allowing the policy to encounter more failure cases on diverse dynamics parameters—proved to be more effective, a result that we attribute to the higher entropy value reached at convergence. Overall, note how DORAEMON is able to consistently maintain the desired in-distribution success rate regardless of the chosen value. This capability can be mostly attributed to the backup optimization problem in (6), as demonstrated by our ablation study in Fig. 13.

## A.4 BEYOND BETA DISTRIBUTIONS

DORAEMON's optimization problem as defined in (4) is not limited to a particular family of parametric distributions. While bounded-support distributions—such as Beta's—are particularly well suited for the Domain Randomization setting, DORAEMON may work with any family whose entropy (for the objective function) and KL-divergence (for the trust region constraint) may be conveniently computed. We therefore compare our method with an implementation that parameterizes the DR distribution as an independent Multivariate Gaussian (i.e. unbounded). Although not generally necessary, we still resample dynamics parameters at training time that fall outside of the search space boundaries defined in Tab. 2[2], in order to fairly compare the results w.r.t. the Beta family. Overall, DORAEMON simply maximizes the entropy of the Gaussian distribution, which is allowed to grow in variance indefinitely. Note how this is in contrast with the KL-divergence term of LSDR's objective, which considers the M-projection formulation. As a result, LSDR optimization problem is analogous to a maximum likelihood objective, rather than a maximum entropy one—assuming target uniforms. We illustrate the comparison of DORAEMON implemented with Beta vs. Gaussian distributions in Fig. 11. The two implementations generally exhibit similar behavior, but may still

---

[2]The Truncated Gaussian implementation of the Scipy library is used for this purpose, as naive consecutive resampling may become increasingly more time consuming as the entropy of the Gaussian increases.

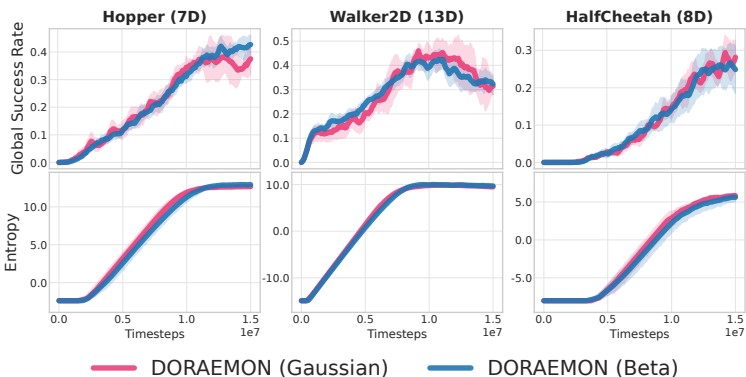

Figure 11: Global success rate computed on the maximum-entropy uniform distribution (top) and entropy of the current training DR distribution (bottom), for distributions parametrized as independent Beta vs. Gaussian. The number of randomized dynamics parameters per environment is reported in parenthesis.

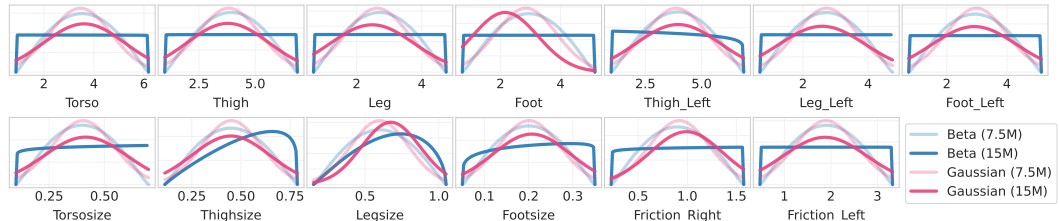

Figure 12: Comparison of Beta vs. Gaussian distributions at different times during training with DORAEMON on the Walker2D environment (training timesteps in parenthesis). When considering fixed feasible boundaries for DR parameters, a Beta distribution proves to be more suitable as it can inherently converge to a uniform distribution over the defined boundaries.

be chosen according to the specific task at hand. In particular, we depict an example of distributions encountered during the training process in Fig. 12, respectively for the two parametric families. Despite showing a similar global success rate, it is likely that the Gaussian parametrization may bias the learner around the mean of the distribution, resulting in poorer generalization towards the corner of the boundaries considered.

## B    CONNECTIONS WITH CURRICULUM LEARNING

The idea of training reinforcement learning agents under moving MDPs isn't confined to the domain randomization setting in sim-to-real problems, and, e.g., has been concurrently explored in the field of curriculum learning (CL). In particular for the latter setting, self-paced algorithms (Klink et al., 2020b;a) are formulated as constrained optimization problems that automatically increase the difficulty of the task—i.e. they find a curriculum over the task space—allowing the agent to progressively learn complex target tasks that would otherwise be unfeasible to solve from scratch. Likewise, Huang et al. (2022) and Klink et al. (2022) recently proposed framing CL as interpolations between a source (initial) and a target task distribution by formulating it as an optimal transport problem. Interestingly, the curriculum learning problem shares similarities with the DR setting, as different simulation parameters may be analogously considered as different tasks—see Sec. 4.1 of the recent survey by Muratore et al. (2022b). This becomes particularly clear when considering curricula over MDPs that share the same state and action space, and only differ by transition dynamics. However, the DR problem importantly departs in that (1) no target task distribution is given, and that (2) the current underlying task is unknown to the agent at test time—real-world dynamics parameters are unknown. In turn, DR asks for agents that must not forget previously learned tasks, and that exhibit generalizable behaviors among latent dynamics parameters.

Despite the different problem formulations and assumptions, DORAEMON draws inspiration from the curriculum learning literature and follows the intuition that more complex tasks may be obtained with increased diversity of dynamics parameters at training time. As a result, DORAEMON is able to cope with the lack of assumptions in the sim-to-real problem w.r.t. a standard CL formulation, while leveraging the flexibility of the self-paced optimization problem for automatically adjusting the DR distribution.

In this section, we therefore summarize the connections between CL and DR settings, and consequently those between self-paced and DORAEMON algorithms. As we break the discussion down into paragraphs for the sake of clarity, we will refer to Fig. 13 to support the stated claims with empirical results.

**Unknown target distribution.** The CL problem assumes that a target task distribution is given, indicating the desired tasks that we are ultimately interested in solving. This fundamentally departs from the DR problem setting, where simulator parameters are randomized to seek generalization to *unknown* real-world task distributions. To cope with this, LSDR proposed considering target *uniform* distributions that should be designed to be as wide as possible, making it a hyperparameter of the algorithm. In contrast, AutoDR removed the dependence on target distributions, and simply attempts to converge to a uniform distribution that approaches infinite support. Overall, the unknown target distribution therefore makes CL algorithms impractical to be applied to the DR setting directly. A common ground between the two settings may still be identified when restricting the problem to bounded domains of dynamics parameters, and assuming uninformative—*i.e.* uniform—target distributions for CL. In such case, both problems would effectively aim to gradually solve all tasks within the predefined boundaries, with no bias towards any region in particular. As this restriction is made throughout our experimental analysis to allow for comparison with LSDR—which requires a target distribution to be defined in the first place—we can directly compare the results of DORAEMON with, e.g., SPDL (Klink et al., 2020b). Nevertheless, non-trivial adaptations of SPDL must still be made to cope with non-observability of dynamics parameters, as clarified in the following paragraph.

**Contextual MDPs vs. Latent MDPs.** The CL problem assumes to work with contextual MDPs (Modi et al., 2018), where an *observable* context variable—namely the underlying task parameters—augments the environment and determines the underlying transition dynamics. This assumption fundamentally differs from a sim-to-real problem, where dynamics parameters are unknown to the agent at inference time. In turn, solving a sim-to-real problem requires finding an optimal context-agnostic policy of the form $\pi(a|s)$, in contrast to the conventional context-conditioned policies learned in CL $\pi(a|s, \xi)$. Domain Randomization may indeed be conveniently formulated by introducing Latent MDPs (LMDPs), as recently proposed by the theoretical study in (Chen et al., 2022). It's worth noting that context-conditioned policies may still be learned—as simulators offer full information—but would end up being impractical for deployment in the real world. Previous work has, *e.g.*, attempted to work around this by inferring dynamics parameters at test time (Yu et al., 2018). However, the more popular approach is to leverage memory-based policies to gain information on the underlying transition dynamics, and therefore implicitly on the latent parameters (see Sec. 9 in Akkaya et al. (2019)). Furthermore, tailored implementation tricks to the sim-to-real problem setting such as *asymmetric actor-critic* have been introduced to additionally ease the training process under partial observability, namely by providing full information to the learned critic networks which are nevertheless *not* queried at test time (Akkaya et al., 2019). As reported in our method Section 4, such implementation details are already included in DORAEMON, but may be easily overlooked when simply comparing our optimization problem to self-paced methods. We therefore report a thorough analysis on the effect of each of these components in Fig. 13, as we show a detailed ablation of DORAEMON from the perspective of SPDL optimization problem. Our empirical evaluation demonstrates the contribution of each added component, highlighting the importance of accurately dealing with the different assumptions induced by the DR problem setting. In particular, we notice that history-based policies (*SPDL + History*) may at times even hinder policy training w.r.t. standard Markovian policies (*SPDL*), and only become particularly effective when combined with asymmetric actor-critic (*SPDL + History + Asymmetric*). Finally, we report an *SPDL Oracle* baseline which reflects the performance of dynamics-conditioned policies $\pi(a|s, \xi)$ as in a contextual MDP setting, despite being of little help to solve our problem. Interestingly enough, such

policies sometimes appear to poorly generalize to out-of-distribution dynamics (see global success rate curves), likely due to overfitting to the dynamics observed at training time.

**I-projection vs. M-projection.** Non-trivial connections among DORAEMON, LSDR, AutoDR and the self-paced methods can also be observed when solely focusing on their optimization problems. As all these methods attempt to either minimize a KL-divergence objective or maximize entropy (implicitly for AutoDR and explicitly for DORAEMON), we attempt to study their differences and similarities from a conceptual and technical viewpoint. Interestingly, self-paced methods rely on the I-projection objective, which effectively becomes equivalent to a maximum entropy objective when considering target uniform distributions. On the other hand, however, the I-projection formulation limits the choice of parametric distributions to be of bounded support that is fully included within the target. This limitation led us to design a novel implementation of SPDL with Beta distributions for the analysis in Fig. 13, departing from the original paper implementation. In contrast, LSDR proposes an M-projection objective for the DR setting, likely to avoid restricting the optimization parameters to bounded distributions when considering uniform targets. By doing so, however, the optimization fundamentally switches to a maximum likelihood objective, providing an easy explanation as to why Gaussian distributions by LSDR do not converge to high entropy values in Fig. 2. Overall, DORAEMON does not rely on a KL-divergence objective, hence both (1) does not need to specify a target uniform and (2) can be implemented with any parametric family (see Sec. A.4 for DORAEMON with Gaussian distributions). In turn, DORAEMON may be considered a hybrid formulation that bridges self-paced methods for the DR setting in sim-to-real, conceptually achieving the best of both worlds. Finally, notice how AutoDR also drops the dependency on the target distribution, but crucially comes with a number of inherent limitations: it's confined to uniform DR distributions only, may only adjust the distribution curriculum with fixed $\Delta$ steps (limited flexibility w.r.t. self-paced methods), and may only update one dimension at a time (inefficient use of training samples).

**Effect of the backup optimization problem.** Lastly, we discuss the contribution of the backup optimization problem defined in Eq. (6), the final distinctive technical novelty in DORAEMON's formulation w.r.t. self-paced methods. Despite the clean theoretical formulation, the practical implementation of SPDL must consider approximations of the performance constraint, which could lead to unfeasible regions of the optimization landscape at the following iteration—namely, experience an in-distribution success rate lower than $\alpha$. This aspect has not been explicitly addressed in prior self-paced works, which simply keep training the current policy when this occurs. However, we suspect this problem to be particularly important to overcome in the DR setting, as it's harder to assume that policy performance would increase as we keep training. Conversely, it's likely that problematic dynamics parameters may even catastrophically affect the learning process. To this end, we propose backtracking the entropy distribution by directly looking for maximum in-distribution success rate within the current trust region. Our empirical analysis in Fig. 13 shows impressive tracking of the $\alpha$ hyperparameter when the backup optimization problem is included (*DORAEMON*), which holds for different environments, different trust region sizes, and different $\alpha$ values (also refer to Fig. 10). Indeed, notice that all other ablations struggle to keep a feasible performance constraint as the entropy of the distribution grows (see in-distribution SR curves entering the red-shaded, unfeasible area). As a side note, we noticed that such addition would not necessarily translate to better global success rates. After further investigation (see Fig. 9), we conclude that, as DORAEMON gradually attempts to find *easier* regions of the dynamics space through backtracking, policies would occasionally suffer from catastrophic forgetting and struggle to retain good performance over previously experienced dynamics. Future work may investigate more sophisticated ways to track the in-distribution success rate, while ensuring that critically useful skills for generalization shall not be forgotten. While out of the scope of this work, multimodal distributions may also be leveraged to isolate problematic regions in the dynamics space, yet maintain generalizability over feasible dynamics.

## C PANDAPUSH TASK

The details of our experimental setting for the PandaPush environment are reported here.

We consider the 7-DoF Franka Panda robotic arm mounted on a table and tasked with pushing a perfectly squared box of size 10cm per side. We analyze the setting where the center of mass of

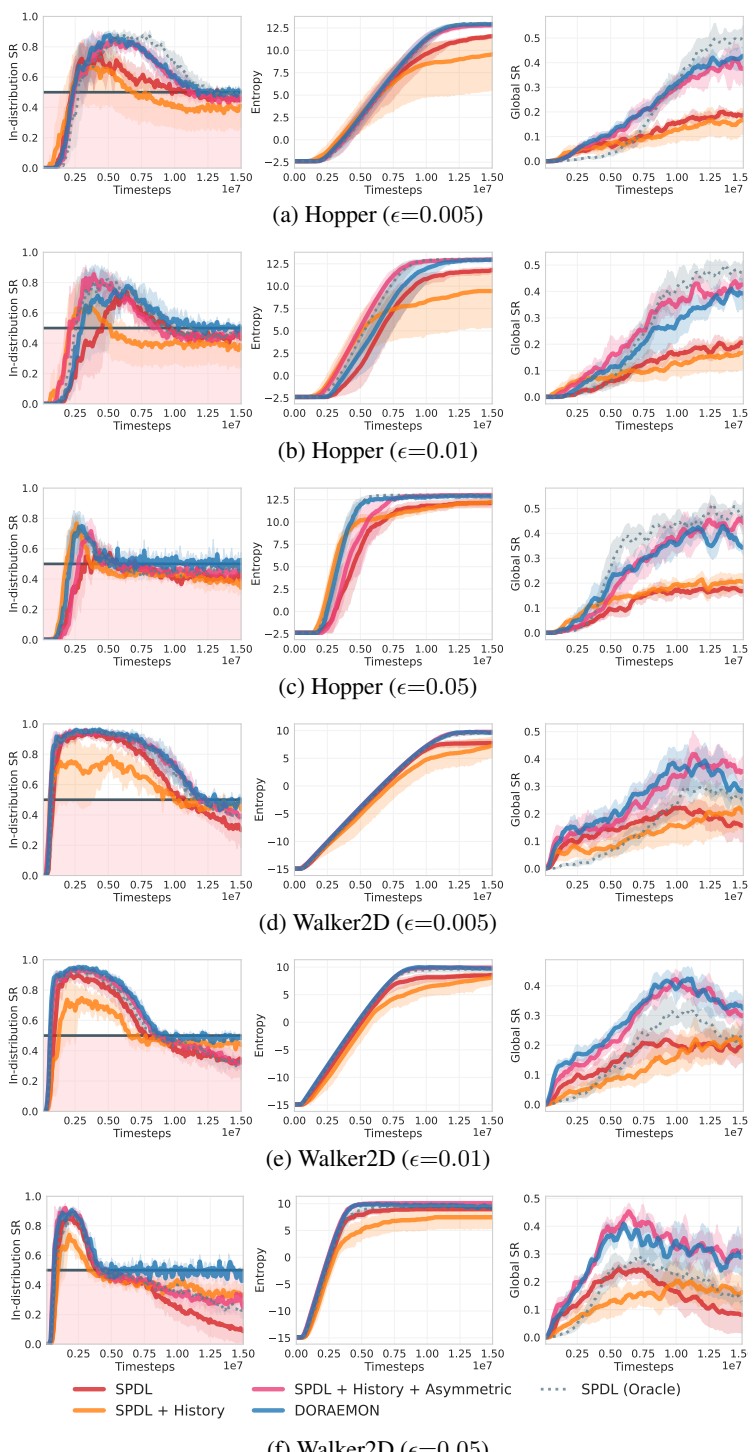

Figure 13: Ablation of DORAEMON from the perspective of self-paced methods in curriculum learning (SPDL). The results demonstrate that all components of DORAEMON are vital to maintain the desired in-distribution success rate $\alpha$, while achieving high generalization. The red-shaded area indicates the region where the performance constraint is violated, i.e. in-distribution success rate lower than $\alpha$. SR stands for *success rate*.

the box is unknown; hence, pushing the box will result in unpredictable behavior. Notably, policies are conditioned on a history of state-action pairs, which may be used as information to compensate for the unknown parameter. Therefore, we aim to learn versatile policies that can solve the task for a varying center of mass through robust (to the reality gap) and adaptive (to dynamics changes) behavior. For the sake of clarity, we break down the task implementation into the following points.

**State space and initialization.** The underlying MDP considers a 20-dim state space, which includes joint positions (7D), joint velocities (7D), box position on the table (2D), box rotation angle $z$ about the z-axis (2D encoded as $sin(z), cos(z)$), and goal position (2D). We initialize the robotic arm as shown in Fig. 15, such that it is already sufficiently close to the box. The initial location of the box is randomized with a uniform noise of $\pm 1cm$, and we further slightly randomize the box height by $10cm \pm 0.005cm$ to regularize the learned behavior against *specification gaming*[3]—*i.e.*, avoid cases where the agent would undesirably rely on the box corners or edges to maximize the return. Finally, we apply a mild Gaussian noise with $\sigma = 0.0011$ to the joint velocity observations—an informed guess based on the noise estimated on the real system.

**Action space and controller.** The agent interacts with the environment by sending joint *acceleration* commands, in normalized space $[-1, 1]$ for each joint. We then denormalize the commands by multiplying for the predefined joint acceleration limits—namely, $[6, 3, 4, 5, 6, 8, 8]$—and integrate the commanded acceleration forward in time to get a desired trajectory to follow, given the current joint position and velocity. We then track such trajectory through a PD controller and a feedforward term—which is trivially computed by multiplying the inertia matrix by the commanded acceleration. Overall, we query the policy at a frequency of $50Hz$, and follow the resulting low-level 20ms-trajectory at $1000Hz$.

**Reward function and success notion.** We reward the agent with the current (negative) squared distance of the box from the goal location, to drive a pushing motion. We further engineer the reward function to encourage adjustments in close proximity of the goal where the squared distance alone would be considerably flat. Therefore, we finally design a reward function of the form $-d^2 - \log(d^2 + 0.01)$, with $d$ being the distance from the target. The obtained reward signal is then rescaled such that it starts at a value of zero, given the initial distance to the target. Finally, we additionally penalize the magnitude of the accelerations commanded by the agent to encourage smooth and safe execution of the policies. For running DORAEMON and AutoDR, we then define a notion of success based on the box distance directly. Indeed, note that a desired lower bound on the return may be cumbersome to define with shaped reward formulations. We therefore denote success for trajectories where the final location of the box falls within a 3cm distance to the goal.

**Domain Randomization.** We randomize a total of 17 dynamics parameters to cross the reality gap in the PandaPush environment. As for the sim2sim experiments, we define a bounded search space over the parameters to keep a notion of a maximum entropy distribution, used for assessing generalizability across the baselines. The boundaries of each randomized parameter are reported in Tab. 3. In particular, we designed them with minimal prior knowledge, and no automatic system identification. Rather, our goal is to train on maximally wide distributions, such that generalization can be achieved even without careful estimation of the real parameters. Notably, we set the starting distribution of LSDR, AutoDR, and DORAEMON to be centered around the values of 0.1 for all joint damping and friction parameters, which is considerable lower than the center of the search space. We observe that higher values of these parameters yield to a higher task difficulty, especially at the early stages of training—*i.e.* higher friction values make it harder for the agent to start collecting rewards from pushing the box. Interestingly, Fixed-DR policies—which are trained over a uniform distribution that spans the whole search space—were unable to learn despite including both low and high friction values.

**Evaluation setting.** We found the PandaPush environment to be more challenging to learn from a pure RL perspective, leading to more unstable training curves w.r.t. the MuJoCo testbed environments. However, good performing policies were still retained by iteratively evaluating the performance at training time over the maximum entropy distribution, and keeping track of the best-

---

[3]Read more on specification gaming at Blog post https://www.deepmind.com/blog/specification-gaming-the-flip-side-of-ai-ingenuity

|  | Parameters | Boundaries |
|---|---|---|
|  | Box mass (1D) | $[0.2, 0.6]$ |
|  | Surface Friction (1D) | $[0.03, 0.5]$ |
| PandaPush (17D) | Joint damping (7D) | $[0.025, 2.5]$ |
|  | Joint friction (7D) | $[0.03, 3]$ |
|  | Center of mass (1D) | $[-0.05, 0.05]$ |

Table 3: Search space boundaries for the 17 dynamics parameters randomized in the PandaPush environment. The center of mass boundaries refer to the shift in centimeters from the geometrical center, along the perpendicular axis of the pushing direction.

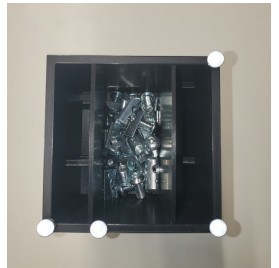

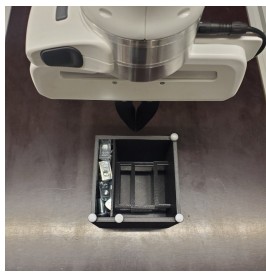

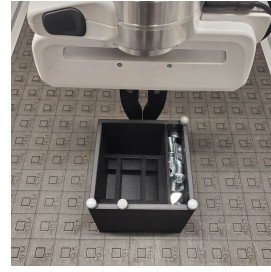

(a) Easy CoM, low friction  (b) Hard CoM, low friction  (c) Hard CoM, high friction

Figure 14: Illustration of the three configuration settings used for evaluating PandaPush policies in the real world. We adjust the center of mass (CoM) by replacing the steel bolts inside the box, and we increase the sliding friction by changing the table top (c).

performing policies by means of highest global success rate. Finally, we compare each baseline by evaluating the average return and success rate of the best-performing policy across 10000 episodes in simulation (1000 episodes for 10 seeds), and 30 rollouts in the real world (3 rollouts for 10 seeds). In particular, we carry out the tests on the real setup by varying the surface friction and center of mass of the box, resulting in 3 test configuration for which we collect one rollout each (see Fig. 14).

**Supplementary results and discussion.** We complete the experimental evaluation of the Panda-Push task by reporting the training curves for all benchmark methods in Fig. 17, averaged over 10 seeds. The training curves reflect the final results in Tab. 1 of the main manuscript, as we observe a much higher performance of DORAEMON's policies. We attribute this to the capability of our optimization problem to efficiently guide the distribution towards high-entropy regions, while maintaining a steady increase of the global success rate. Interestingly, the performances seem to degrade significantly when the maximum-entropy distribution is approached (see DORAEMON curves in Fig. 17 at timesteps 3.5M). In this scenario, DORAEMON finds it hard to quickly recover from such occurence, opening avenue for future work directions to mitigate such behavior. We then investigate the moving Beta distribution of DORAEMON during training, and illustrate the history of encountered distributions in Fig. 16 for a subset of 5 parameters. We find that, while our method converges to a near maximum-entropy uniform (yellow curve), best generalization is achieved for an intermediate entropy distribution (green curve). This is analogous to the case of the Walker2D and Swimmer tasks in Sec. 5.2. We remind that, regardless of the drop in performance due to increased diversity of the dynamics parameters, for all baselines we keep track of the best-performing policy while training by means of global success rate, and use this one for real-world evaluation.

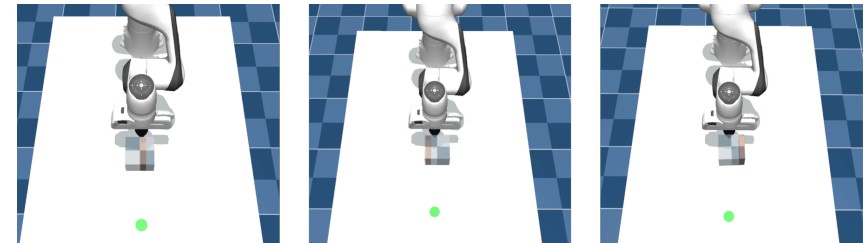

Figure 15: Three random configurations of the PandaPush setup in simulation: the center of mass (CoM) of the box is randomized along the axis perpendicular to the pushing direction of the box. This makes it a particularly interesting setup for studying the adaptivity of the learned history-based policies. The green dot highlights the goal location.

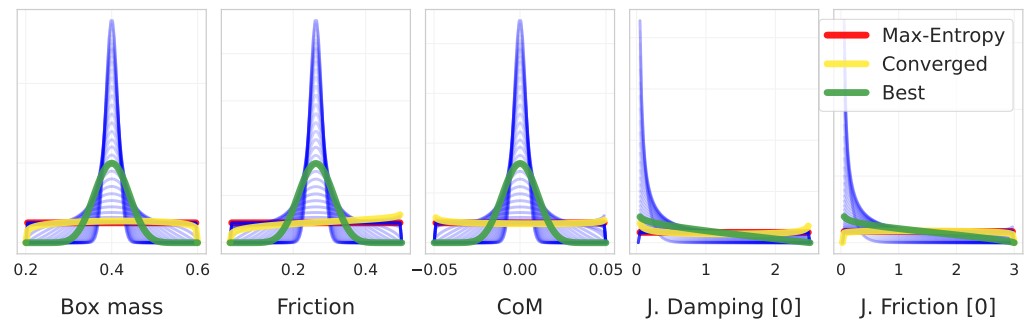

Figure 16: DORAEMON's Beta moving distribution across five representative dynamics parameters of the PandaPush task (blue curves are more opaque for more recent iterations). The final converged distribution (yellow) and the best distribution by means of global success rate (green) are also reported. "CoM", "J. Damping [0]", and "J. Friction [0]" abbreviate Center of Mass, first-joint damping, and first-joint friction respectively.

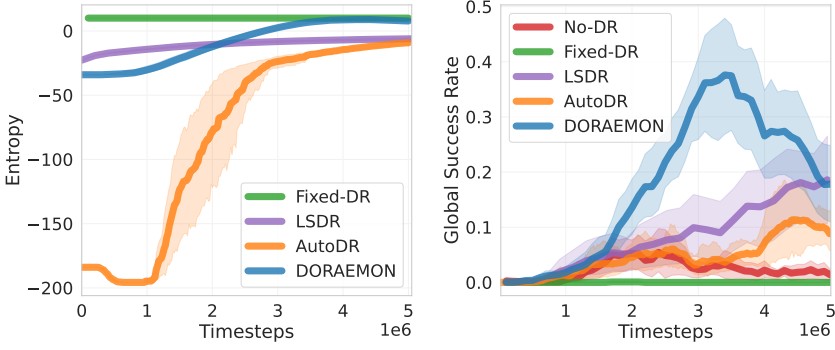

Figure 17: PandaPush sim-to-sim results: global success rate computed on the maximum-entropy uniform distribution (right) and entropy of the current training DR distribution (left).

