# OpenReview forum: "Domain Randomization via Entropy Maximization"
_ICLR.cc/2024/Conference — ICLR 2024 poster_

### Official Review · Reviewer_7kiW · 2023-10-26

**Soundness:** 2 fair
**Presentation:** 3 good
**Contribution:** 2 fair
**Rating:** 5
**Confidence:** 3

**Summary:**

This paper presents a new domain randomization method that tries to overcome the performance and generalization gap by maximizing the entropy of the distribution of dynamic parameters while retaining certain success probability during training. The authors conduct simulation experiments as well as sim-to-real experiments.

**Strengths:**

- The article presents its content in a clear and concise manner.
- The method exhibits novelty and has been well formalized.

**Weaknesses:**

- The baselines utilized in this study appear to be somewhat outdated, which raises the question of whether more recent advancements in domain randomization have been considered. It is highly recommended that the authors explicitly address this concern by providing a specific clarification on the existence of any updated domain randomization approaches, and it would be beneficial for the authors to incorporate additional baseline experiments that encompass these newer methodologies.
- While the method proposed in this study demonstrates a certain level of innovation, it does not appear to be exceptionally groundbreaking. Therefore, it is imperative to present more compelling experimental results. Conducting additional simulation experiments, as well as sim-to-real experiments, would be highly recommended.

**Questions:**

- In Table 1, the accuracy of the Fixed-DR approach is notably low, prompting the need for the authors to provide further explanations, if not overlooked, in order to clarify any potential factors contributing to this outcome.

---

> ### Author Response · Authors · 2023-11-20
> **Rebuttal (1/2)**
>
> We highly appreciate the reviewer's feedback.
>
> - > The baselines utilized in this study appear to be somewhat outdated, which raises the question of whether more recent advancements in domain randomization have been considered
>
> We have carefully revised the literature during this work, and we are particularly confident on the existing works in the field of Domain Randomization (DR) for sim-to-real transfer.
> Despite DR being a popular approach to cross the reality gap in robotics, the vast majority of papers still carry out a "Fixed-DR" approach (often simply called UDR)---i.e. uniform boundaries are manually engineered in a back-and-forth tuning to obtain desirable performances. This likely stems from (1) the lack of public code implementations (as in the case of AutoDR), or (2) the requirement of real-world data for inference-based DR methods. Related works that follow the latter direction have been thoroughly described in Sec. 2, but crucially depart from our setting where no real-world data is available.
>
> Particularly, a thorough survey has been published just last year by F. Muratore et al. [2], carefully describing the landscape of DR approaches in literature. The survey importantly distinguises approaches that make use of real-world time series data for parameter inference (Adaptive Domain Randomization), from approaches that keep a static DR distribution (Static Domain Randomization).
>
> In absence of real-world data, AutoDR has been considered the state-of-the-art method since its release, and it has been adopted just last year by NVIDIA [1] as their go-to algorithm for sim-to-real transfer. In this newer work [1], they also still compare AutoDR with manual DR (or Fixed-DR), and demonstrate the benefits of training on a progressively wider distribution by AutoDR. As a side note, the work in [1] proposed a more efficient parallelized version of AutoDR w.r.t. the original method, and adopted manually-tuned specific step sizes $\Delta$ for each parameter dimension. As these changes simply either require more tuning or higher-end compute, we stick to the comparison with the original AutoDR method in our paper.
>    While it received less attention, LSDR was published in IROS 2020 and proposed a novel approach to optimize DR distributions without using real-world data. However, LSDR departs from DORAEMON and AutoDR in that the M-projection of the KL-divergence is considered---yielding a maximum likelihood objective instead of a maximum entropy one---and it requires expensive Monte Carlo policy evaluations to step the current distribution.
>
> Overall, we kindly ask the reviewer to point out any newer work that has been overlooked in our work, as, to the best of our knowledge, we already compared to all most recent baselines in the field.
>
>
> - > While the method proposed in this study demonstrates a certain level of innovation, it does not appear to be exceptionally groundbreaking. Therefore, it is imperative to present more compelling experimental results. Conducting additional simulation experiments, as well as sim-to-real experiments, would be highly recommended.
>
> In our work, we have shown that DORAEMON is able to outperform all baselines in all tasks (in terms of maximum global success rate) and crucially demonstrates an impressive zero-shot sim-to-real transfer in our complex pushing task. In addition, DORAEMON has fewer hyparameters than the other methods (see Sec. A.2).
> Nevertheless, we have conducted several more experiments to provide novel useful insights on the performance of DORAEMON. In particular, we added:
>   1. **A novel sensitivity analysis of DORAEMON in Sec. A.2**;
>   2. **A novel Section A.3** to further discuss the trade-off between the in-distribution success rate vs. entropy;
>   3. **A novel Section A.4** with an implementation of DORAEMON with Gaussian distributions, instead of Beta's used in the main manuscript.
>   4. **A novel ablation analysis of DORAEMON in Fig. 13**, demonstrating the effect of the backup optimization problem in (6);
>
> We claim DORAEMON will open interesting and promising future work directions, which can ultimately provide "exceptionally groundbreaking" results in the field. Such directions include, e.g., the combination of a maximum entropy objective like DORAEMON's with inference-based methods such as SimOpt [3] or DROPO [4]. This way, policies can be learned to best generalize across dynamics parameters located around high-likelihood regions, which can be inferred from real-world data.
>
> We believe that our detailed explanations and additional results clarify the contribution of our approach and we hope that the reviewer would consider them.

---

> ### Author Response · Authors · 2023-11-20
> **Rebuttal (2/2)**
>
> - > In Table 1, the accuracy of the Fixed-DR approach is notably low, prompting the need for the authors to provide further explanations, if not overlooked, in order to clarify any potential factors contributing to this outcome.
>
> The usual way of approaching domain randomization with fixed (static) distributions is extremely sensitive to the choice of the uniform boundaries. This is a well-known problem in literature, and is the main motivation for works like ours, AutoDR, and LSDR. In particular, choosing uniform boundaries that are too wide, notiously leads to performance collapse, as the learner can not deal with this much variability in the dynamics of the environment (i.e. a form of over-regularization). Interestingly, we noticed that progressively growing the training distribution as in DORAEMON, this effect is largely mitigated---e.g. compare the results of DORAEMON with Fixed-DR when the entropy converges to the maximum. This finding is in line with the experimental analysis in AutoDR's work [5]. In particular for the Pushing task case mentioned by the reviewer, almost all Fixed-DR policies were completely unable to learn any meaningful behavior, and ended up moving very slowly or simply staying still. Refer to Fig. 11 in the appendix for the full sim-to-sim results in the PandaPush task. As stated in Sec. C, we found that higher friction parameters would significantly hinder the training process, and primarily attribute this phenomenon to the poor results of Fixed-DR policies.  Moreover, as the reward function includes a penalty for aggressive policy actions, Fixed-DR policies may end up learning a "do nothing" policy as a suboptimal way to deal with the severe randomization.
>
>
>
> [1] Handa, A., et al. "DeXtreme: Transfer of Agile In-hand Manipulation from Simulation to Reality." arXiv (2022).
>
> [2] Muratore, Fabio, et al. "Robot learning from randomized simulations: A review." Frontiers in Robotics and AI (2022): 31.
>
> [3] Chebotar, Yevgen, et al. "Closing the sim-to-real loop: Adapting simulation randomization with real world experience." 2019 International Conference on Robotics and Automation (ICRA). IEEE, 2019.
>
> [4] Tiboni, G. et al. "DROPO: Sim-to-real transfer with offline domain randomization." Robotics and Autonomous Systems 166 (2023): 104432.
>
> [5] Akkaya, Ilge, et al. "Solving rubik's cube with a robot hand." arXiv preprint arXiv:1910.07113 (2019).

---

> > ### Comment · Reviewer_7kiW · 2023-11-22
> > **Thanks for the response**
> >
> > Thank you for the detailed explanation, which has addressed my initial concerns to some extent. I am willing to raise the score accordingly.

---

### Official Review · Reviewer_TxSB · 2023-10-28

**Soundness:** 3 good
**Presentation:** 3 good
**Contribution:** 3 good
**Rating:** 6
**Confidence:** 3

**Summary:**

The paper proposes a novel approach to addressing the "sim-to-real transfer" challenge in reinforcement learning. The presented approach, DORAEMON, aims to maximize the diversity of dynamics parameters during training by incrementally increasing randomness while ensuring a sufficiently high success probability of the current policy. This results in highly adaptable and generalizable policies that perform well across a wide range of dynamics parameters.

**Strengths:**

A straightforward method to explore the environment dynamics parameters is proposed for RL algorithms to enhance their generalization.

The method is simple, and its effectiveness is demonstrated in the toy example and the experiments.

**Weaknesses:**

The method is heuristic and lacks theoretical analysis.

**Questions:**

1.	The proposed method calls the RL algorithm to update the policy for every dynamics parameters sampling, which might lead to an inefficient algorithm. Maybe the embedded RL algorithm only need to return a relatively approximate solution? Does this works for DORAEMON. This is expected to be made clear.

2.	The authors adopt univariate beta distributions for \nu_{\phi}, which might simplify problem (6). But for more general distributions, solving (6) might be challenging. Maybe this should be further investigated. Or is it the case that some commonly used distributions, e.g. Gaussian, are effective and meanwhile make (6) tractable. Or some variational methods can be adopted.

---

> ### Author Response · Authors · 2023-11-20
>
> We highly appreciate the reviewer's feedback.
>
>
> - > The method is heuristic and lacks theoretical analysis.
>
>    We agree with the Reviewer that the paper lacks theoretical analysis. As the primary goal of our work is to present a novel solution to the sim-to-real problem, we focus on a strong empirical analysis to highlight the relevance of our method to the field.
>    In particular, while a theoretical analysis is not provided, DORAEMON introduces a technically sound optimization problem that works in a more principle manner than the previous algorithmic heuristic proposed by AutoDR---e.g., AutoDR can only work with uniform distributions and simply steps the distribution with fixed $\Delta$ increments.
>    In our latest revision, we also provide a number of new experiments to further complement the experimental evaluation:
>    - Added a sensitivity analysis w.r.t. hyperparameters $\epsilon$ (Sec. A.2), and $\alpha$ (Sec. A.3);
>    - Added experiments on DORAEMON with Gaussian distributions, instead of Beta's (Sec. A.4);
>    - Added an ablation analysis of DORAEMON (Fig. 13), and discussion on its connections to curriculum learning (Sec. B);
>
>    We believe that these new analyses, together with the already thorough experimentation of the method in 6 sim2sim environments and a complex sim2real task, make our contribution highly relevant. A detailed theoretical analyses is left for a journal extension of our work.
>
>
>
> - > The proposed method calls the RL algorithm to update the policy for every dynamics parameters sampling, which might lead to an inefficient algorithm. Maybe the embedded RL algorithm only need to return a relatively approximate solution?
>
> DORAEMON does not change anything about the underlying RL subroutine, which, in fact, is not even "informed" about the change of the DR distribution. This allows to use any RL algorithm of choice, and simply call DORAEMON's optimization problem in Eq. (4) every time new $K$ trajectories have been collected by the training agent (line 3 of Algorithm 1). We can interpret this as a standard RL procedure where the learning agent is progressively presented with MDPs $\in \mathcal{U}$ of varying transition dynamics. Therefore, we shall not worry about the intermediate approximate solutions in between each iteration, as DORAEMON automatically adjusts the distribution updates according to the goodness of the current agent. In other words, the distribution won't be updated as drastically (less entropy increase, if any) when the agent only barely satisfies the minimum desired success rate, and viceversa (this phenomenon can be clearly seen in our newly introduced analysis in Fig. 10).
>
>
> - > The authors adopt univariate beta distributions for \nu_{\phi}, which might simplify problem (6). But for more general distributions, solving (6) might be challenging. Maybe this should be further investigated. Or is it the case that some commonly used distributions, e.g. Gaussian, are effective and meanwhile make (6) tractable. Or some variational methods can be adopted.
>
> DORAEMON can work with any family of parametric distributions, as long as the entropy (for the objective function) and the KL-divergence (for the trust region constraint) can be conveniently computed. In our experiments, we use independent Beta distributions, and therefore deal with $2$ parameters per dimensions to be optimized by our problem in (4) and (6). This results in a number of optimized parameters of $4$, $14$, $26$, $16$, $14$, and $34$ respectively for the CartPole, Hopper, Walker2D, HalfCheetah, Swimmer, and the PandaPush environments (see Tab. 2 and Tab. 3).
> Overall, we find our optimization problem to be particularly efficient. We computed statistics on $500$ runs of our optimization problem in the Hopper environment, and observed an average duration of $4.19 \pm 1.72$ seconds. In particular, we make use of the Scipy library and its convenient "Trust Region Constrained Algorithm"---see line 897 of `doraemon/doraemon.py` in our anonymous codebase in the supplementary material.
>    Finally, we tested DORAEMON with Gaussian distributions and **reported the results in a novel Appendix Section A.4**. This analysis further demonstrates that our optimization problem is just as tractable with other parametric distributions, and also led to similar results.

---

> > ### Comment · Reviewer_TxSB · 2023-11-22
> >
> > Thank the authors for your response!

---

> ### Author Response · Authors · 2023-11-22
>
> We thank the Reviewer for acknowledging our rebuttal. We kindly ask whether our rebuttal has been helpful for the Reviewer to consider improving their assessment of our work, and we are glad to provide more clarifications if needed.

---

### Official Review · Reviewer_W7po · 2023-10-31

**Soundness:** 3 good
**Presentation:** 3 good
**Contribution:** 3 good
**Rating:** 6
**Confidence:** 5

**Summary:**

The paper introduces domain randomization via entropy maximization, a constrained optimization framework that directly maximizes the entropy of the training distribution while retaining generalization capabilities. The authors empirically evaluate their method in several simulated control environments. Additionally, they successfully showcase zero-shot transfer in a robotic manipulation task under unknown real-world parameters, emphasizing its practical applicability.

**Strengths:**

1. The proposed framework utilizes entropy maximization to gradually enlarge the randomization range, which is sound.
2. Experiments compared to control results demonstrate the effectiveness of the proposed methods.
3. The robot manipulation experiments indicate that the proposed method has the potential for use in real-world tasks.

**Weaknesses:**

1. Only Beta distributions are considered in the experiments. It is encouraged to add more distribution types to the experiments.
2. Additional visualizations need to be included to illustrate the trade-off between performance and entropy. For example, in Figure 2, the performance of Walker2D and Swimmer decreases when the entropy increases. It is important to explore the relationship between the randomized variable and performance, and to determine the range within which performance decreases.
3. Adding more real-world experiments is encouraged. PushCube is a relatively easy task in robot manipulation.

**Questions:**

Can this domain randomization method potentially be applied to object randomness? For example, in ManiSkill environments, some tasks include variations in objects. Can we use maximum entropy to gradually learn from these different objects?

---

> ### Author Response · Authors · 2023-11-20
> **Rebuttal (1/2)**
>
> We thank the reviewer for taking the time to evaluate our work and providing insightful comments.
>
> - > Only Beta distributions are considered in the experiments. It is encouraged to add more distribution types to the experiments.
>
> While we chose Beta distributions as they can better capture the probability mass of the target uniform distribution, in principle we can use any parametric distribution whose entropy (for the objective function) and KL-divergence (for the trust region constraint) can be computed straightforwardly. In response to the reviewer's comment,   we implemented DORAEMON with Gaussian distributions and report the results in the **newly added Section A.4**. We also depict a comparison of converged distributions in the two cases (Beta's and Gaussian's) in Fig. 12.
>
>
> - > Additional visualizations need to be included to illustrate the trade-off between performance and entropy.
>
> We provide more details on the trade-off between the performance of DORAEMON as the entropy increases, in our novel Appendix sections. In particular, we provide a thorough investigation on multiple environments (including Walker2D as suggested by the reviewer) measuring the in-distribution success rate, the entropy, and the global success rate for varying values of $\epsilon$ and $\alpha$. Such analysis sheds light on the effect of the backup optimization problem in maintaining a feasible performance constraint, and how this does not always translate to a better global success rate.
>   Overall, we complemented the original manuscript with:
>   1. **A novel sensitivity analysis in Sec. A.2** where we particularly show the effects of the backup optimization problem in Fig. 8 and Fig. 9 for changes of $\epsilon$;
>   2. **A novel Section A.3** to discuss the trade-off between in-distribution success rate vs. entropy for the Walker2D environment (Fig. 10), as suggested by the reviewer;
>   3. **A novel ablation analysis of DORAEMON in Fig. 13**, demonstrating that all combinations that do not incorporate a backup optimization problem may not maintain a feasible performance constraint during training;
>
>   In particular, we point out the results in Fig. 10 for the Walker2D environment, where DORAEMON steadily maintains a feasible in-distribution success rate during training, regardless of the choice of the hyperparameter $\alpha$. However, we observed that a sound tracking of the in-distribution success rate does not necessarily translate to better global success rate, which still shows a decreasing trend. We investigate this phenomenon in Fig. 9, depicting a pair of distributions found at different times during training: despite seemingly having similar entropy values according to the plot, the backup optimization problem significantly changes the distribution parameters to maintain feasible performance constraints, and in turn, moves to easier regions of the dynamics landscape. As a result, the global success rate is also affected. We further ablate the presence of the backup optimization problem in Fig. 13, and discover that its contribution does not negatively affect the global success rate, which would decline regardless if we simply kept training the policy when the performance constraint is violated (i.e. as in SPDL baselines).
>
>
> - > Adding more real-world experiments is encouraged. PushCube is a relatively easy task in robot manipulation.
>
> While adding more real-world experiments would certainly be beneficial to limit-test the algorithm, we believe our pushing task to be a particularly well representative testbed for our analysis, and likely more complicated than it looks at first glance. Indeed, it is easy to overlook the complexity of the task from the Domain Randomization point of view: while a simple pushing-cube task would not be considered particularly challenging in RL literature, it becomes immediately more troublesome when testing under unknown real-world parameters (e.g., center of mass, frictions), hence requiring the policy to generalize over multiple dynamics. This is easily reflected by the poor performance of an agent that randomizes parameters too heavily (Fixed-DR), and by the lack of generalization for agents that do not randomize parameters at all (No-DR).
>
>   Given the complexity of setting up real world experiments, and the respective design of a representative simulation environment in the limited rebuttal period, we postpone the further investigation of DORAEMON in a variety of real-world manipulation environments for a journal extension of this work.

---

> ### Author Response · Authors · 2023-11-20
> **Rebuttal (2/2)**
>
> - > Can this domain randomization method potentially be applied to object randomness? For example, in ManiSkill environments, some tasks include variations in objects. Can we use maximum entropy to gradually learn from these different objects?
>
> We thank the reviewer for this comment. As we have already stated in the final sentences of Sec. 4.1 (*"[...] the formulation is not restricted to a particular family of parametric distributions or even continuous random variables–e.g., discrete distributions over object shapes could be used."*), it is indeed possible to apply DORAEMON to random object shapes. For example, one could consider a categorical distribution over $N$ object types, with initial low entropy---i.e. most of the probability mass is assigned to a single object, and the rest is spread across the others. Then, DORAEMON can be used to progressively increase the entropy of the distribution, e.g. parametrizing it through a softmax of N continuous parameters. Ultimately, DORAEMON will attempt to converge to a uniform distribution over all N object shapes, while ensuring desirably high performance along the process.
>
> Overall, this is a rather interesting point, and opens the possibility to test DORAEMON on discrete dynamics distributions beyond object shapes. We leave this as an open direction for future work.

---

> > ### Comment · Reviewer_W7po · 2023-11-21
> > **Thanks for the response!**
> >
> > Thank you for your comprehensive experiments and clarifications. My primary concerns have been addressed. However, the practicality and challenges of applying this method, especially in challenging robot manipulation tasks, remain uncertain without further experimentation. The authors should consider applying the method to manipulation tasks if the paper is not accepted this time.

---

> > > ### Author Response · Authors · 2023-11-22
> > >
> > > We thank the reviewer for the quick response and for considering our novel experiments and clarifications.
> > >
> > > We are extremely interested in applying DORAEMON to more challenging manipulation tasks, as we believe the method will be an important contribution for the solution of sim-to-real problems, especially for dynamics-sensitive tasks.
> > > As they are crucially time-consuming, we plan to apply DORAEMON to more sim-to-real experiments in its appropriate journal extension. For example, we plan on considering tasks with unknown obstacle shapes, and contact-rich dynamics with deformable objects.
> > >
> > > We hope the reviewer will support the current conference version that already proves strong evidence of state-of-the-art performances in six sim2sim tasks and one sim2real task---a novel 17-dynamics parameter DR cube-pushing task whose codebase is released to the public.

---

> ### Comment · Reviewer_W7po · 2023-11-22
> **Final response**
>
> Generally, simple control experiments are not impressive these days, as algorithms can overfit these simple settings. That's why I am asking for robot manipulation experiments, which can demonstrate the usefulness of your method. I think the direction is generally good, but more realistic and convincing experiments are necessary to achieve better scores for this paper in ML conferences.
>
> **To AC: The paper is still broadline without more experiments over realistic settings.**

---

> > ### Author Response · Authors · 2023-11-22
> >
> > We totally understand the Reviewer's opinion and thank them for the overall assessment of our work.

---

### Official Review · Reviewer_QW63 · 2023-11-01

**Soundness:** 3 good
**Presentation:** 3 good
**Contribution:** 3 good
**Rating:** 6
**Confidence:** 3

**Summary:**

The paper introduces DORAEMON, which revisits domain randomization from the perspective of entropy maximization. Specifically, instead of maximizing the expected total return over the distribution of the dynamics parameters, this paper proposes a constrained optimization problem that directly maximizes the entropy of the dynamics distribution subject to a constraint on the success probability.  Based on this formulation, this paper proceeds to offer an algorithmic implementation that decouples this maximization into two subroutines: (1) Update the policy by any off-the-shelf RL algorithm under the current dynamics parameter; (2) Under the current policy, update the dynamics parameter to improve the entropy with the help of a KL-based trust region. Accordingly, a toy experiment is provided to demonstrate the dynamics distribution that DORAEMON converges to. The proposed algorithm is then evaluated on both sim-to-sim (MuJoCo) and sim-to-real tasks (PandaPush) against multiple baseline DR methods.

**Strengths:**

- The method introduced in this paper is quite intuitive and reasonable in concept and avoids some inherent issues of DR. Specifically, as the standard DR requires a pre-configured fixed prior distribution over the support of the environment parameter (which would require some prior domain knowledge), the proposed DORAEMON framework learns to maximize the entropy of dynamics distribution and hence naturally obviates this issue. (That said, in the meantime, the threshold needed for defining a successful trajectory also requires some domain knowledge, but probably a bit less)
- The proposed algorithm is evaluated in a variety of domains (including both sim-to-sim and sim-to-real scenarios), and the empirical results demonstrate quite promising performance of the DORAEMON framework (in terms of success rate).
- The paper is well-written and very easy to follow, with justification and explanation whenever needed in most places.

**Weaknesses:**

Overall I could appreciate the proposed reformulation of DR, but there are some concerns regarding the algorithm:

- DORAEMON appears to be conceptually very similar to the AutoDR, or ADR in the original paper (Akkaya et al., 2019). They both define some custom indicators of success and iteratively increase the entropy of the dynamics distribution. With that said, DORAEMON appears to be yet a somewhat different implementation of the idea highlighted by ADR.

- Based on the above, while the two approaches arise from similar ideas, DORAEMON appears to have a better success rate across almost all environments. It is not immediately clear whether the performance improvement comes from which specific part of the design or it is just a matter of different choices of hyperparameters. While there is a one-sentence discussion on the authors’ conjecture in Section 5.2 (about the potential data inefficiency), it is expected to have a deeper dive into the root cause of this performance difference.

- The successful rate for certain tasks, e.g., Walker in Figure 2 and PandaPush in Figure 11, decline after reaching maximum entropy. However, the algorithm does not dynamically reduce the entropy in response to a decrease in the success rate, which might be necessary for maintaining performance consistency. This appears not consistent with the objective in (4). As the discussion in Section 5.2 does not fully address this phenomenon, more explanation would be needed.

- Another concern lies in the constraint based on the success rate. Specifically, the use of success rate largely ignores the effect of the poor trajectories, which could be arbitrarily poor and degrade the robustness of the learned policy. By contrast, in the standard DR, the objective is to consider the expected total return over all the possible trajectories. As the experimental results reported in the paper all focus on the “success rate”, the robustness issue is thereby largely neglected.

**Questions:**

Detailed comments/questions

- In practice, is it computationally easy to optimize (4)? The constrained problem does not seem to be convex (even under Beta distribution)?

- Could the authors specify the alpha values for Figure 2 and Figure 11? When the entropy matches the max entropy, the global success rate aligns with the local success rate. If the alpha is set to 0.5, why does the global success rate drop below 0.5 when entropy is at its maximum?

- How to design the entropy jointly for multiple dynamics parameters? (For example, simply taking the product of multiple univariate distributions like AutoDR?)

- In Section 3: The notation of reward function shall be consistent (mathcal or not)

------------------ Post-rebuttal ------------------

I would like to thank the authors for the detailed response. Most of my questions have been addressed, especially the ablation study. That being said, regarding the comparison of DORAEMON and AutoDR, while I can understand authors' response on their differences, these differences still appear more like subtle implementation choices, and this keeps me from giving a higher rating.

---

> ### Author Response · Authors · 2023-11-20
> **Rebuttal (1/2)**
>
> We thank the reviewer for their valuable feedback. We further discuss the raised concerns by the reviewer.
>
> - > It is not immediately clear whether the performance improvement comes from which specific part of the design or it is just a matter of different choices of hyperparameters. While there is a one-sentence discussion on the authors’ conjecture in Section 5.2 (about the potential data inefficiency), it is expected to have a deeper dive into the root cause of this performance difference.
>
>   As the reviewer readily pointed out, DORAEMON builds on the same intuition as AutoDR to progressively increase the entropy of the training distribution, but follows a drastically different algorithm to do so. In particular, let us break down the fundamental differences between DORAEMON and AutoDR:
>   1. **Uniforms distributions only**: by definition, AutoDR is limited to a training distribution parametrized as uniform. This is a major limitation of the algorithm, which may not capture correlations, nor multimodal effects, nor unbounded dynamics parameters. In contrast, DORAEMON's optimization problem can work with any family of parametric distributions. To support this claim with further evidence, **we added experiments on DORAEMON with Gaussian distributions in the newly introduced Section A.4**.
>   2.  **Fixed step size**: AutoDR may only step the distribution boundaries by a fixed step size $\Delta$, which considerably limits the flexibility of the algorithm. This resulted in a much higher variance of the entropy curves across seeds (see Fig. 2). Instead, DORAEMON leverages the solution of an explicit optimization problem to step the current distribution while remaining within a trust region in KL-divergence space. In turn, DORAEMON can dynamically step all dimensions of the distribution as much as needed to maintain a feasible performance constraint. A recent work by NVIDIA [1] implemented AutoDR with per-dimension $\Delta$ values, which attempts to provide more flexibility at the expense of drastically more hyperparameter tuning.
>   3.  **Inefficient use of training samples**: AutoDR does not leverage all training data as information to update the distribution. Rather, it alternates pure training samples from the uniform distribution (50%) to dynamics parameters sampled at the boundaries (50%). While all are used to update the current policy, only the latter samples provide information on the success rate at the boundary to step the distribution. Moreover, these samples may even be discarded if the success rate is not sufficiently high to increase the entropy of the distribution---see $CLEAR(D_i)$ instruction in Alg. 1 of [2]. Conversely, DORAEMON uses all training samples of the current iteration to step the distribution (never discards them), and does not bias the sampling process to occur at the boundaries half the time.
>   4.  **More domain knowledge for backtracking**: AutoDR introduces an additional low return threshold $t_L$ to shrink the entropy of the distribution when performances are too low. This mechanism is analogous to the backup optimization problem introduced by DORAEMON, despite our method not requiring an additional hyperparameter, and relying on a more flexible, explicit optimization (for the same reasons as in the **"Fixed step size"** point). The original paper [2] simply sets $t_L$ to be half the high-return threshold $t_H$, but we suspect that this hyperparameter should likely be tuned with task-specific domain knowledge.
>
> - > The successful rate for certain tasks, e.g., Walker in Figure 2 and PandaPush in Figure 11, decline after reaching maximum entropy. However, the algorithm does not dynamically reduce the entropy in response to a decrease in the success rate, which might be necessary for maintaining performance consistency.
>
>   We agree that the limited explanation in the main manuscript makes it hard to fully understand this phenomenon, despite we tried our best to deliver the maximum content given the page limit.
>
>   In our updated manuscript, and in response to the reviewer's comment, we delve into the details of this phenomenon in the Appendix:
>   1. **We added a novel sensitivity analysis in Sec. A.2** where we particularly show the effects of the backup optimization problem in Fig. 8 and Fig. 9 for changes of $\epsilon$;
>   2. **We added a novel Section A.3** to discuss the trade-off between in-distribution success rate vs. entropy for the Walker2D environment (Fig. 10), as suggested by the reviewer;
>   3. **We added an ablation analysis of DORAEMON in Fig. 13**, demonstrating that all combinations that do not incorporate a backup optimization problem may not maintain a feasible performance constraint during training;

---

> ### Author Response · Authors · 2023-11-20
> **Rebuttal (2/2)**
>
> In particular, we point to the results of Fig. 10 for the Walker2D environment: notice how DORAEMON can steadily maintain a feasible in-distribution success rate during training, regardless of the choice of the hyperparameter $\alpha$. However, we observed that a sound tracking of the in-distribution success rate does not necessarily translate to better global success rate, which still shows a decreasing trend. We investigate this phenomenon in Fig. 9, depicting a pair of distributions found at different times during training: despite seemingly having similar entropy values according to the plot, the backup optimization problem significantly changes the distribution parameters to maintain feasible performance constraints, and in turn, moves to easier regions of the dynamics landscape. As a result, the global success rate is also affected. We further ablate the presence of the backup optimization problem in Fig. 13, and discover that its contribution does not negatively affect the global success rate, which would decline regardless if we simply kept training the policy when the performance constraint is violated (i.e., as in SPDL baselines).
>
> - > Another concern lies in the constraint based on the success rate.
>
> We believe this point to essentially be a matter of design choice. Our formulation makes use of the success rate to allow for custom success notions to be defined, hence more flexibility. For example, consider the inclined plane task in Sec. 4.2: a succesful trajectory is defined as such when the cart is balanced around the center of the plane for longer than 25 timesteps. An average (expected) return formulation would be impractical in this case. Furthermore, a success notion allows the designer to encode task-specific toleration to errors in the problem.
> Overall, DORAEMON may still be used with success notion that are simply defined as return threshold (as in our sim-to-sim analysis in Sec. 5.2), which would be equivalent to consider a *median* return formulation as opposed to the *average* return formulation suggested by the reviewer. This likely makes DORAEMON's optimization problem less affected by catastrophic returns (which would not affect the median performance), which we argue could be beneficial. Nevertheless, the importance sampling estimator in Eq. (5) can be easily adjusted to consider an average return performance constraint.
>
> - > In practice, is it computationally easy to optimize (4)? The constrained problem does not seem to be convex (even under Beta distribution)?
>
> DORAEMON is particularly efficient. We computed statistics on $500$ runs of our optimization problem (4), and observed an average duration of  $4.19 \pm 1.72$ seconds. As we run (4) iteratively $150$ times for each training session---$15$M timesteps, with $100000$ training timesteps in between each iteration---this results in approx. $10$ minutes of computational power, w.r.t. a total training time of about $16$ hours.
>   In particular, we make use of the Scipy library and its convenient "Trust Region Constrained Algorithm"---see line $897$ of `doraemon/doraemon.py` in our anonymous codebase in the supplementary material.
>
> - > Could the authors specify the alpha values for Figure 2 and Figure 11? When the entropy matches the max entropy, the global success rate aligns with the local success rate. If the alpha is set to 0.5, why does the global success rate drop below 0.5 when entropy is at its maximum?
>
>  We use $\alpha=0.5$ across all our experiments. The only parameter that is specifically tuned for each environment is the trust region size (see Sec. A.2 for more information).
>   Please refer to our comments above in this post for a detailed explanation on why the global success rate may decrease despite the entropy **appears** to be at its maximum.
>
>
> - > How to design the entropy jointly for multiple dynamics parameters? (For example, simply taking the product of multiple univariate distributions like AutoDR?)
>
> Correct. As we work with univariate Beta distributions, the product of these PDFs is considered for the joint distribution whose entropy needs to be computed. We then simply sum the univariate distribution entropies---a convenient property of the differential entropy when dealing with independent distributions, which can be easily verified.
>
> - > In Section 3: The notation of reward function shall be consistent (mathcal or not)
>
> Thank you for spotting the typo. We fixed the notation.
>
> [1] Handa, A., et al. "DeXtreme: Transfer of Agile In-hand Manipulation from Simulation to Reality." arXiv (2022).
>
> [2] Akkaya, Ilge, et al. "Solving rubik's cube with a robot hand." arXiv  (2019).

---

### Official Review · Reviewer_eC78 · 2023-11-03

**Soundness:** 3 good
**Presentation:** 2 fair
**Contribution:** 1 poor
**Rating:** 5
**Confidence:** 5

**Summary:**

Domain Randomization (DR) is a common technique used to reduce the gap between simulations and reality in Reinforcement Learning (RL), which involves changing dynamic parameters in simulations. The effectiveness of DR, however, largely depends on the chosen sampling distribution for these parameters. Too much variation can regularize an agent's actions but may also result in overly conservative strategies if the parameters are randomized too much. This paper introduces a new method for enhancing sim-to-real transfer, dubbed DOmain RAndomization via Entropy MaximizatiON (DORAEMON). DORAEMON is a constrained optimization framework that aims to maximize the entropy of the training distribution while also maintaining the agent's ability to generalize. It accomplishes this by incrementally expanding the range of dynamic parameters used for training, provided that the current policy maintains a high likelihood of success. Experiments show that DORAEMON outperforms several DR benchmarks in terms of generalization and showcase application in a robotics manipulation task with previously unseen real-world dynamics.

**Strengths:**

The paper is well-written and easy to follow. The authors also conducted real-world robotics experiments.

**Weaknesses:**

* Limited technical novelty. The formulation (Eq. 4) is very similar to that of SPRL[1], SPDL[2], CURROT[3], and GRADIENT[4]. Setting the target distribution to be uninformative -- uniform distribution could transform their objectives into something very similar to Eq.4, and they do not necessarily converge to the final target distribution.
* Beta distribution is often not a reasonable choice. It cannot handle multi-modal distribution, while many existing works can handle arbitrary empirical distributions [1,2,3,4].
* Given the similarity to the existing work as discussed in the first point, the authors should compare DORAEMON to them.
* Missing related work:
    * Klink, Pascal, et al. "Curriculum reinforcement learning via constrained optimal transport." International Conference on Machine Learning. PMLR, 2022.
    * Huang, Peide, et al. "Curriculum reinforcement learning using optimal transport via gradual domain adaptation." Advances in Neural Information Processing Systems 35 (2022): 10656-10670.
    * Cho, Daesol, Seungjae Lee, and H. Jin Kim. "Outcome-directed Reinforcement Learning by Uncertainty & Temporal Distance-Aware Curriculum Goal Generation." arXiv preprint arXiv:2301.11741 (2023).

Ref:
[1] Klink, Pascal, et al. "Self-paced contextual reinforcement learning." Conference on Robot Learning. PMLR, 2020.

[2] Klink, Pascal, et al. "Self-paced deep reinforcement learning." Advances in Neural Information Processing Systems 33 (2020): 9216-9227.

[3] Klink, Pascal, et al. "Curriculum reinforcement learning via constrained optimal transport." International Conference on Machine Learning. PMLR, 2022.

[4] Huang, Peide, et al. "Curriculum reinforcement learning using optimal transport via gradual domain adaptation." Advances in Neural Information Processing Systems 35 (2022): 10656-10670.

**Questions:**

See weaknesses.

---

> ### Author Response · Authors · 2023-11-20
> **Rebuttal (1/2)**
>
> Thank you for taking the time to evaluate our work.
>
> > Limited technical novelty. The formulation (Eq. 4) is very similar to that of SPRL[1], SPDL[2], CURROT[3], and GRADIENT[4]. Setting the target distribution to be uninformative – uniform distribution could transform their objectives into something very similar to Eq.4, and they do not necessarily converge to the final target distribution.
>
>
> The primary goal of the paper is to present a novel solution to the sim-to-real problem, and our empirical results make DORAEMON highly promising and relevant for the field. As a matter of fact, our method outperforms AutoDR, the currently considered state-of-the-art method for sim-to-real transfer in absence of real-world data---recently adopted by NVIDIA in [1] as their go-to DR method for sim-to-real transfer, well after self-paced methods have been introduced. To achieve such results, DORAEMON **does take inspiration from the self-paced optimization problem in the curriculum setting, but our method is *far* from a straightforward application of self-paced methods in a sim-to-real problem**, in contrast to what the reviewer's comment might reflect.
> To prove this, we implemented SPDL and compared it to DORAEMON in a thorough analysis that adds individual (and combinations of) components to SPDL, clearly shedding light on the differences between the two algorithms (see Fig. 13). The results demonstrate that a naive application of self-paced methods is impractical or even impossible in the domain randomization setting, and that a number of technical novelties---history-based policies, asymmetric actor-critic, and a backup optimization problem---has to be introduced to obtain desirable performance and prevent violation of the performance constraint.
>
> Overall, we report a summarized list of the key aspects where our sim-to-real problem formulation departs w.r.t. curriculum learning (CL) settings (see Sec. B for details):
> - **Unknown target distribution**: the knowledge of a target distribution is a strict requirement for CL methods. Setting uninformative uniform targets could be an option to bridge its formulation to a sim-to-real problem. However, this prompts the user to define the uniform boundaries as a hyperparameter of the method: while the boundaries should be designed to be as wide as possible, SPDL's practical implementation would likely suffer from uniforms that approach infinite support. This also holds for LSDR, which by definition requires a target uniform distribution to be defined. In contrast, DORAEMON drops such dependency.
> - **Contextual MDPs vs. LMDPs**: CL methods such as SPDL work in a contextual RL setting where a context variable that represents the current task is in principle observable and available to the learner---e.g. goal states that progressively get harder to reach. Conversely, the DR formulation induces the solution of a Latent MDP [2], as dynamics parameters may not be directly observed. Such theoretical change effectively leads to fundamental performance differences for the learning agent (see Fig. 13).
> - **I-projection vs. M-projection**: While SPDL cannot be directly applied to a sim-to-real problem due to the different assumptions, it shares similarities with both DORAEMON and LSDR optimization problems. Interestingly, we observe how SPDL makes use of the I-projection formulation to move the training distribution towards the target, while LSDR proposes the M-projection counterpart. In turn, SPDL may drive distributions towards maximum entropy when considering target uniforms (but is limited to parametric families of bounded support for the optimization), while LSDR can work with unlimited support distributions (but results in a maximum likelihood objective that would not grow the entropy indefinitely). Overall, DORAEMON does not rely on a KL-divergence objective, hence (1) can converge to maximum entropy uniform distributions, and (2) can be implemented with any parametric family, including unbounded ones (see the new Section A.4 for DORAEMON with Gaussian distributions).
> - **Backup optimization problem**: the practical implementation of self-paced methods does not allow for the entropy to decrease along the process. In turn, the rise of violated performance constrains is not explicitly addressed, and the agent simply keeps training in such occurrences. Our novel backup optimization problem introduced in DORAEMON overcomes this problem. The new empirical analysis in Fig. 13 demonstrates the effectiveness of DORAEMON to consistently move in the feasible region of the optimization landscape, whereas SPDL quickly suffers from training performance collapse.
>
> While not in-depth, these differences have also been highlighted by the recent survey on DR for sim-to-real transfer by F. Muratore et. all [3], in their Sec. 4.1.

---

> ### Author Response · Authors · 2023-11-20
> **Rebuttal (2/2)**
>
> We thank the reviewer for giving us the chance to add such in-depth discussion in the paper, which would likely be helpful for future readers.
> Furthermore, we are currently doing the best we can to run more experiments as the rebuttal period goes on. In particular, we plan on complementing the analysis in Sec. B for the remaining sim-to-sim environments.
>
> Finally, note that GRADIENT and CURROT, despite trying to solve the same problem as SPDL, work on particle-based distributions and bring significant technical novelties in their optimization problems---namely, framing it as an optimal transport problem. While this could serve as inspiration to potentially find extensions of DORAEMON that analogously frame it as an optimal transport problem, this is certaintly out of the scope of this work and left as a future work direction.
>
>
>
> #### Beyond Beta distributions
>
> > Beta distribution is often not a reasonable choice. It cannot handle multi-modal distribution, while many existing works can handle arbitrary empirical distributions [1,2,3,4].
>
> DORAEMON can work with any family of parametric distributions for which their entropy (for the objective function) and KL-divergence (for the trust region constraint) may be conveniently computed, just like SPDL.
> In fact, **we tested DORAEMON with Gaussian distributions and report the results in the newly added Section A.4.**
>
>
> We believe that our detailed explanations and additional results clarify the contribution of our approach and we hope that the reviewer would consider them for a thorough re-assessment of our work.
>
>
> #### Missing related works
>
> > Missing related work: [...]
>
> We included a citation of the aforementioned works in our latest revision: CURROT and OUTPACE in Sec. 2, and GRADIENT in Sec. B.
>
>
>
> [1] Handa, A., et al. "DeXtreme: Transfer of Agile In-hand Manipulation from Simulation to Reality." arXiv (2022).
>
> [2] Chen, Xiaoyu, et al. "Understanding Domain Randomization for Sim-to-real Transfer." ICLR (2021).
>
> [3] Muratore, Fabio, et al. "Robot learning from randomized simulations: A review." Frontiers in Robotics and AI (2022): 31.

---

> > ### Comment · Reviewer_eC78 · 2023-11-21
> > **RE:**
> >
> > Thank the authors for the detailed response. However, my concerns regarding the technical novelty (w.r.t self-paced RL) and limitation induced by the parametric family still hold.
> >
> > - First, I think beta distribution is also bounded, and you need to choose the boundaries as well. I understand that the proposed method also works with unbounded parametric distribution if KL and entropy can be computed efficiently, in principle. However, the original choice of distribution is not convincing.
> >
> > - Second, it is not obvious to me why "Such theoretical change (LMDP vs. CMDP) effectively leads to fundamental performance differences for the learning agent." I think the authors could provide more theoretical justification, or maybe the authors could give more detailed insight into Chen, Xiaoyu, et al.
> >
> > - Third, as discussed in the rebuttal, being a parametric family of distribution is not necessarily an advantage. The added experiment with Gaussian distributions does not address my concern: it is still uni-modal, and any assumptions on the underlying distribution may limit the flexibility of the proposed method and thus the final effects.

---

> > > ### Author Response · Authors · 2023-11-22
> > > **Rebuttal pt. 2 (1/2)**
> > >
> > > We thank the reviewer for the quick response.
> > >
> > > - > However, the original choice of distribution is not convincing.
> > >
> > >    We compare our method with LSDR and Fixed-DR baselines that by definition require a fixed target uniform distribution, as they aim to achieve the best generalization on it. As it wouldn't be fair to assess their generalization capabilities on distributions different than their corresponding target, we therefore benchmark all methods in terms of the success rate achieved on the same target distribution, including those methods which do not necessarily require boundaries (AutoDR and DORAEMON). This is done in order to diminuish the number of external factors affecting the comparison. In other words, all methods can only sample parameters within the target distribution, making it a much fairer comparison. Furthermore, having a reference distribution for generalization makes it possible for us to benchmark the methods in the first place: how would we compute the ability of the methods to generalize across dynamics otherwise? A target, reference, distribution needs to be defined for the sake of comparison, and to allow for different methods to move within the same space of dynamics parameters.
> > >    For this reason, DORAEMON has originally been tested with Beta distributions, and AutoDR---which can only work with uniform distributions---is prevented from growing the boundaries wider than the target distribution in our implementation.
> > >    Beyond this analysis, we then demonstrate that (unbounded) Gaussian distributions may also be used for DORAEMON and achieve the same performances as in the Beta implementation (see Fig. 11), highlighting that the method's superior performance does not result from the particular choice of the distribution.
> > >    Overall, our thorough empirical analysis clearly shows that (1) our method can in principle work with any family of parametric distributions (and in particular, Beta's and Gaussian's), and that (2) this particular design choice did not significantly affect the results in our case, attributing the superior performance to the remaining components of the algorithm. We indeed remind that LSDR is also implemented with Gaussian distributions.
> > >
> > > - > Second, it is not obvious to me why “Such theoretical change (LMDP vs. CMDP) effectively leads to fundamental performance differences for the learning agent.”
> > >
> > >    We are happy to provide more details on this, as we took the effort to write an in-depth novel Section B with a thorough ablation analysis and comparison with self-paced methods, in response to the reviewer's concerns.
> > >    The contextual RL setting in curriculum learning often provides observable variables that determine the task difficulty, hence one may train policies conditioned on the task parameters explicitly. For example, a curriculum over goal states allows for policies to be trained with input knowledge on the current goal. The partially-observable setting of Domain Randomization, however, completely prohibits this: dynamics parameters shall be considered latent variables of the underlying distribution of MDPs, and are unobservable and unknown at test time. This problem setting has been recently formulated with the introduction of Latent MDPs in [3]. We provide clear empirical evidence of this problem in Fig. 13: the SPDL-Oracle policies $\pi(a|s,\xi)$ are conditioned on the sampled dynamics parameters $\xi$, while SPDL reflects the performance of a simple Markovian policy $\pi(a|s)$. The evident performance difference between the two approaches, demonstrates that a naive application of self-paced methods to Domain Randomization would perform incredibly poorly. We then discuss, based on recent theoretical studies [3], how Latent MDPs can be dealt with by conditioning policies on the history of state-action pairs previously experienced. This allows policies to get information on the dynamics of the current MDP, and act accordingly, despite not knowing the dynamics parameters directly (i.e. a form of implicit system identification can occur). Despite the sound thoretical ground provided by [3], our experiments in Fig. 13 yet do not show significant improvements when adding history alone. We attribute this phenomenon to the higher complexity of the state space, which now considers a much higher number of dimensions (we consider 5-timestep histories).

---

> > > ### Author Response · Authors · 2023-11-22
> > > **Rebuttal pt. 2 (2/2)**
> > >
> > > Overall, we find that the best combination can be achieved when adding both history and the asymmetric actor-critic paradigm. Unfortunately, such integrations to SPDL still do not ensure that policies move within a feasible landscape of the constrained optimization problem---see how the in-distribution success rate in Fig. 13 falls in the unfeasible region below $\alpha$, and keep a steady decreasing trend for all SPDL baselines. We therefore notice how the backup optimization problem defined in Eq. (6) allows DORAEMON to both retain generalization capabilities and maintain the desired performance threshold with impressive consistency across multiple environments, multiple trust region sizes, and multiple values $\alpha$ (see Fig. 10 for the latter claim).
> > >
> > >
> > >
> > > - > Third, as discussed in the rebuttal, being a parametric family of distribution is not necessarily an advantage. The added experiment with Gaussian distributions does not address my concern: it is still uni-modal, and any assumptions on the underlying distribution may limit the flexibility of the proposed method and thus the final effects.
> > >
> > >    The reviewer opens a concern about the limited flexibility of DORAEMON. Yet, DORAEMON demonstrates that it can work with any family of parametric distributions, which is far better than the current state-of-the-art method AutoDR that is limited to Uniform distributions **only**. LSDR, on the other hand, could work with any parametric family, but it has only been shown with Gaussian distributions or simple discrete distributions.
> > >    Furthermore, the reviewer statement *"Beta distribution is often not a reasonable choice. It cannot handle multi-modal distribution, while many existing works can handle arbitrary empirical distributions [1,2,3,4]."* is technically wrong: [1] and [2] make the same parametric assumptions on the distribution as DORAEMON and, as they require the minimization of a KL-divergence objective, a closed-form solution is likely needed to perform the optimization. In turn, this makes [1] and [2] impractical for multi-modal distributions for which the KL cannot be computed in closed-form (e.g. mixtures of Gaussians) or approximated easily. In fact, for multi-modal distributions for which KL computation is possible, Doraemon could readily be used as well.
> > >    To conclude, the reviewer essentially converges to the point that DORAEMON is not mature enough because multi-modal distributions may not be captured, despite all considered baselines in the field completely disregard this limitation, or even have stricter assumptions than us (i.e. AutoDR). While more flexible representations could provide benefits to the algorithm, it is unclear why this point is presented as a primary weakness to the method: DORAEMON introduces a technically sound constrained optimization problem with strong empirical evidence that superior sim-to-real transfer can be obtained vs. state-of-the-art methods, while allowing for more distribution flexibility than current methods in the field, and fewer hyperparameters---rendering it a highly relevant contribution for sim-to-real problems.
> > >
> > >
> > > Ref:
> > >
> > > [1] Klink, Pascal, et al. "Self-paced contextual reinforcement learning." Conference on Robot Learning. PMLR, 2020.
> > >
> > > [2] Klink, Pascal, et al. "Self-paced deep reinforcement learning." Advances in Neural Information Processing Systems 33 (2020): 9216-9227.
> > >
> > > [3] Chen, Xiaoyu, et al. "Understanding Domain Randomization for Sim-to-real Transfer." ICLR (2021).

---

> > > > ### Comment · Reviewer_eC78 · 2023-11-23
> > > > **Re:**
> > > >
> > > > Thank the authors for the detailed response. I will raise my score one step for now while digesting other reviews and engaging in discussions in the next stage.

---

> ### Author Response · Authors · 2023-11-23
>
> We thank the Reviewer for the thoughtful consideration of our rebuttal.

---

### Official Review · Reviewer_BXPC · 2023-11-06

**Soundness:** 4 excellent
**Presentation:** 2 fair
**Contribution:** 4 excellent
**Rating:** 8
**Confidence:** 3

**Summary:**

This paper introduces DORAEMON, a novel domain randomization technique in reinforcement learning, designed to enhance policy generalization across varied environment dynamics. DORAEMON strategically increases the entropy of training distributions, conditioned on achieving a probability of success threshold and by ensuring updates to the entropy are constrained. The aim is to balance both the entropy of the dynamics parameters distribution and task proficiency. Empirical tests across OpenAI Gym benchmarks and on a real-world robotic task highlight DORAEMON's superior adaptability to diverse dynamic settings compared to conventional domain randomization approaches and also demonstrate how the success rate threshold and success definition (i.e., lower bound return threshold) impact performance. This work convincingly demonstrates the potential benefit of applying DORAEMON to systems where sim-to-real policy transfer is important.

**Strengths:**

Originality
The paper introduces a novel and innovative approach to domain randomization. This method is an advancement in the field of RL, not only in the scope of domain randomization but potentially in areas of research outside of domain randomization (e.g., meta-RL). The method's innovation stems from its entropy maximization technique that enables a policy to generalize across a broader range of dynamics, while ensuring that the entropy of the dynamics parameter distribution grows in a manner that does not compromise the policy's probability of success.

Quality
The proposed algorithm, DORAEMON, has been tested in OpenAI Gym benchmarks and in a sim-to-real robotics task, generally demonstrating its superior generalization in comparison to existing domain randomization techniques.

Clarity
The paper has a clear definition of success and the presentation of the results are well-structured. The mathematics foundations of the paper are clear and sound. The figures and empirical results support the authors claims of the superiority of their method in comparison to traditional domain randomization methods.

Significance
This paper is significant to the development of autonomous systems and has the potential for real-world application in industry.

Summary
DORAEMON stands out as an original, high-quality research work with significant implications for both theory and application in reinforcement learning and robotics.

**Weaknesses:**

The paper presents empirical tests across OpenAI Gym benchmarks and a real-world task. However, there may be a need for more diverse environmental tests to fully understand the limits of DORAEMON's generalization ability. For instance, testing in environments with higher-dimensional state spaces or more complex dynamics could provide a more comprehensive picture of the algorithm's robustness.

The research presented is certainly complex and holds significant value to the field. However, some sections of the text may benefit from further clarification to enhance the paper's accessibility to a broader audience. In particular, the density of technical terms and concepts could be balanced with more detailed explanations or simplified language. This could potentially include adding definitions, or providing more background information for non-expert readers. Such revisions would likely make the paper's contributions even more impactful and ensure that a wider range of readers can fully grasp the innovative work you have presented.

**Questions:**

How sensitive is DORAEMON to its hyperparameters (e.g., trust region size, trajectories per distribution update, definition of success, etc.), and what process was followed to select them? Did you perform a sensitivity analysis?

Are there potential negative impacts of DORAEMON that should be discussed, especially regarding its application to real-world systems? While it is beyond the scope of this paper, have you considered the safety of the sim-to-real policy in the real-world robotics task? How does it compare to previous domain randomization methods?

---

> ### Author Response · Authors · 2023-11-20
>
> We highly appreciate the reviewer's time in evaluating our work and providing valuable feedback. In the following, we address the concerns raised by the reviewer.
>
>
> > For instance, testing in environments with higher-dimensional state spaces or more complex dynamics could provide a more comprehensive picture of the algorithm’s robustness.
>
>
> The OpenAI benchmark, and in particular the considered tasks, feature highly nonlinear dynamics, and altering the various dynamics parameters of each task (see Table 2) leads to significantly challenging problems. This can seen by the failure cases of Fixed-DR throughout our experimental analysis (see Fig. 2 and Table 1), which sometimes generalizes even worse than a policy trained with no randomization (see No-DR vs Fixed-DR in Fig. 11). Moreover, our sim-to-real task allows us to test various dynamics properties of a contact-rich manipulation task (e.g. the box response to contacts under variable centers of mass). Note that we will provide the public codebase with the details of our PandaPush setup for the community to use as a novel sim-to-real benchmark.
>
> While more environments would be beneficial for providing more evidence of the effectiveness of DORAEMON, we deem that the current experiments already highlight the benefits of this approach both in sim2sim and sim2real applications.
>
> >  In particular, the density of technical terms and concepts could be balanced with more detailed explanations or simplified language. This could potentially include adding definitions, or providing more background information for non-expert readers.
>
> While we did our best to clearly explain each concept and experiment in the main manuscript, we agree that the content may generally feel a bit tight to fit into 9 pages. In our revision, we made significant additions to the Appendix section in order to add more detailed explanations on the algorithm's behavior (see novel sections A.3, A.4, and B).
>
> > How sensitive is DORAEMON to its hyperparameters (e.g., trust region size, trajectories per distribution update, definition of success, etc.), and what process was followed to select them? Did you perform a sensitivity analysis?
>
> Based on the reviewer's comment, we **complemented Section A.2 in the Appendix with a sensitivity analysis of the trust region size $\epsilon$** (see Fig. 8), as empirical evidence of the process that we followed to select DORAEMON's hyperparameter. In particular, we tune the value of $\epsilon$ individually for each environment, out of a predefined set of 5 candidates in $\{0.1, 0.05, 0.01, 0.005, 0.001\}$. Likewise, we perform the same search for the baseline methods AutoDR and LSDR, respectively, for their trust region size equivalents (please refer to A.2 for a detailed explanation on the selection process). \
>   The sensitivity analysis demonstrates that DORAEMON is considerably robust to the choice of $\epsilon$, i.e., it is able to maintain the desired in-distribution success rate even for large trust region sizes. However, the value of $\epsilon$ highly affects the pace of the growing distribution, therefore, we suspect that the optimal value may change from task to task. \
>   Finally, we also complemented the original sensitivity analysis on the $\alpha$ hyperparameter, by **adding new experiments for the Walker2D and Halfcheetah environments in the new Section A.3**.
>
> > Are there potential negative impacts of DORAEMON that should be discussed, especially regarding its application to real-world systems? While it is beyond the scope of this paper, have you considered the safety of the sim-to-real policy in the real-world robotics task? How does it compare to previous domain randomization methods?
>
> While DORAEMON trains policies to solve a task on a maximally diverse distribution of dynamics, that are no guarantees that real-world dynamics will also be captured. However, in absence of real data and no prior knowledge on real-world dynamics, we claim that DORAEMON would likely be the best go-to option, representing an important step to solve the sim-to-real problem, which is yet to be fully solved. In particular, future work may focus on complementing DORAEMON with inference-based methods, e.g., collecting data from the real world for inferring a maximum entropy distribution that is located around high-likelihood regions of the dynamics space.

---

### Author Response · Authors · 2023-11-20

We appreciate the time taken by all reviewers in evaluating our work, and their insightful comments that allowed us to complement our work with additional experimental analyses, to further highlight the contribution of our work.

In the following, we list the changes in the paper, while in the individual responses, we address each reviewer's comment in detail.

- Fixed notation typo (R -> $\mathcal{R}$) in Sec. 3;
- Added dimensionality of randomized dynamics parameters in Fig. 2;
- Novel sensitivity analysis on the trust region size $\epsilon$, in Sec. A.2;
- Novel Section A.3: detailed analysis of the In-distribution Success Rate vs. Entropy trade-off;
- Novel Section A.4: ablation of DORAEMON with Gaussian distributions, instead of Beta distributions;
- Novel Section B: deep technical and empirical discussion on the connections of DORAEMON with self-paced methods in curriculum learning;
- Added three missing related works in Sec. 2 and Sec. B, as suggested by Reviewer eC78


All changes in the text are highlighted in purple, for the sake of clarity.

---

### Meta-Review · Area_Chair_XSry · 2023-12-10

**Metareview:**

This paper focuses on domain randomization for simulation to real-world (sim2real) transfer. The issue addressed is the balance between the variation of the simulation and the performance of the policy. The proposed approach addresses this by explicitly optimizing for this balance. Extensive experimentation, including a real-world robotics one, is provided to validate the performance of the proposed approach.

Strengths:

- The problem is clearly stated, making the work easily accessible.

- The proposed approach is novel, intuitive, and easily explained

- The set of experiments provided includes an illustrative toy example, several baseline comparisons, sim2sim experiments, as well as a sim2real experiment.

Weaknesses:

- The similarity of the approach to AutoDR is pointed out by the reviews. The paper should clearly describe the differences between the two approaches.

Overall, this work provides clear insights and an intuitively appealing solution with a balanced set of experiments.

**Justification For Why Not Higher Score:**

The paper could have gotten a higher score with a more diverse set of experiments, specifically with real-world robots, as requested by some of the reviewers as well.

**Justification For Why Not Lower Score:**

The paper clearly meets the bar of publication for this venue with several positive points and a few weaknesses.

---

### Decision · Program_Chairs · 2024-01-16

Accept (poster)